# Sources of volatile organic compounds and policy implications for regional ozone pollution control in an urban location of Nanjing, East China

Qiuyue Zhao[1,2], Jun Bi[1*], Qian Liu[2], Zhenghao Ling[3*], Guofeng Shen[4], Feng Chen[2], Yuezhen Qiao[2], Chunyan Li[2], Zongwei Ma[1]

[1]State Key Laboratory of Pollution Control and Resource Reuse, School of the Environment, Nanjing University, Nanjing 210023, China
[2]Jiangsu Key Laboratory of Environmental Engineering, Jiangsu Academy of Environmental Sciences, Nanjing 210036, China
[3]School of Atmospheric Sciences, Sun Yat-sen University, Guangzhou 510275, China
[4]College of Urban and Environmental Sciences, Peking University, Beijing 100871, China
*Correspondence to:* Jun Bi (jbi@nju.edu.cn) and Zhenhao Ling (lingzhh3@mail.sysu.edu.cn)

**Abstract.** Understanding the composition, temporal variability, and source apportionment of volatile organic compounds (VOCs) is necessary for determining effective control measures to minimize VOCs and its related photochemical pollution. To provide a comprehensive analysis of VOC sources and their contributions to ozone ($O_3$) formation in the Yangtze River Delta (YRD) - a region experiencing highest rates of industrial and economic development in China, we conducted a one-year sampling exercise using a thermal desorption-GC (gas chromatography) system for the first time at an urban site in Nanjing (JAES site). Alkanes were the dominant group at the JAES site, contributing ~53% to the observed total VOCs, followed by aromatics (~17%), acetylene (~17%), and alkenes (~13%). We identified seasonal variability in TVOCs with maximum and minimum concentrations in winter and summer, respectively. A morning and evening peak and a daytime trough were identified in the diurnal VOCs patterns. We identified VOC sources using positive matrix factorization and assessed their contributions to photochemical $O_3$ formation through calculating the $O_3$ formation potential (OFP) based on the mass concentrations and maximum incremental reactivities of VOCs. The PMF model identified five dominant VOC sources, with highest contributions from diesel vehicular exhausts ($34 \pm 5\%$), followed by gasoline vehicular exhausts ($27 \pm 3\%$), industrial emissions ($19 \pm 2\%$), fuel evaporation ($15 \pm 2\%$) and biogenic emissions ($4 \pm 1\%$). The results of the OFP calculation inferred that VOCs from industrial and vehicular emissions were found to be the dominance precursors for OFP, particularly the VOC species of xylenes, toluene and propene, which top priorities should be given to the alleviation of photochemical smog. Our results therefore highlight that priority should be given to limited VOC sources and species for effective control of $O_3$ formation in Nanjing.

## 1. Introduction

Volatile organic compounds (VOCs) are key precursors of $O_3$ and secondary organic aerosols (SOA) - a major component of fine particulate matter ($PM_{2.5}$). VOCs significantly contribute to the formation of photochemical

smog, atmospheric oxidative capacity, visibility degradation, and global climate (Jenkin and Clemitshaw, 2000;
Seinfeld and Pandis, 2006), and some VOCs are also known to be toxic to human health. Therefore, in recent
years, much research has focused on the impacts of VOCs due to their influence on atmospheric chemistry and
impacts on human health (Shao et al., 2009 and references therein).

The Yangtze River Delta (YRD) region (Shanghai-Jiangsu-Zhenjiang region) is one of the fastest growing
regions in China, having recently undergone rapid urbanization and industrialization. Rapid economic growth
has led to increased photochemical smog and elevated concentrations of ground-level $O_3$ and fine particulate
matter ($PM_{2.5}$). These conditions have been listed as the most important sources of pollution affecting the
population in the YRD region, and are likely caused by increasing concentrations of VOCs. Therefore, it has
been suggested that controlling VOC emissions is necessary for the effective alleviation of photochemical smog
(Wang et al., 2009; Zhang et al., 2009; Cai et al., 2010; Kurokawa et al., 2013; Ding et al., 2016).

To further understand VOC characteristics and to develop effective policies towards lowering VOC emissions,
a number of sampling campaigns have been conducted to investigate the components, mixing ratios,
photochemical reactivity and emissions of VOCs over the YRD region (Cai et al., 2010; An et al., 2014; Mo et
al., 2015; Pan et al., 2015; Shao et al., 2016; Xu et al., 2017). For example, based on continuous observation data
collected from March, 2011 to February, 2012, An et al. (2014) identified clear seasonal VOC variability in an
industrial area of Nanjing, with maximum and minimum levels observed in summer and winter, respectively.
VOC variability was also found to be strongly influenced by industrial emissions. In contrast, Mo et al. (2017)
found no difference in VOC chemical compositions between residential, industrial and suburban areas of the
coastal industrial city, Ningbo. By comparing the emission-based profiles and those extracted from the positive
matrix factorization (PMF) model, the petrochemical industry was identified as the highest contributor of
ambient VOCs due to the unique industrial structure of Ningbo, which is a coastal city located on the southern
wing of the Yangtze River Delta with petrochemical industry as its lead industry (Mo et al., 2015, 2016). Pan et
al. (2015) conducted emissions measurements of open biomass burning in the rural area of the YRD region and
examined the major contributors to $O_3$ pollution using a box model together with the Regional Atmospheric
Chemical Mechanism. Overall, these studies were conducted in industrialized and/or rural areas of the YRD
region and demonstrate the contribution of industrial emissions and biomass burning towards ambient VOC
levels and their contributions to $O_3$ formation. However, VOC studies in urban areas of the YRD region are
limited and could help to improve our understanding of the spatial variability of VOCs and their environmental
impact, particularly as stricter policies on VOCs and/or photochemical smog have been implemented since 2013
(Fu et al., 2016). Furthermore, the sampling resolution and sampling duration of these studies were relatively
low as the samples were collected using canisters. High-resolution VOC datasets can provide more detailed
information on the temporal and spatial variability, source apportionments, and impact factors of VOCs.

In this study, we collected continuous one-year observational VOC data at an urban site in Nanjing in the YRD
region. The seasonal and diurnal characteristics of VOCs were investigated, and their sources were identified
and quantified using the PMF model. Furthermore, we used a box model together with a Master Chemical
Mechanism (MCM) (version 3.2) to identify the $O_3$-precursor relationships and the contributions of VOC sources
to photochemical $O_3$ formation. Our results were compared with VOCs data from other Chinese megacities.
Based on these findings, we summarize and propose control strategies to minimize VOCs pollution and assess
their implications for Nanjing and the wider YRD region. The results provide useful information towards
lowering photochemical pollution in the YRD region as well as other regions in China.
**2. Methodology**
**2.1. Sampling campaign**
We continuously measured VOC concentrations from January to December, 2016, at an observation station on
the rooftop of an office building (~80 m above the ground level) of the Jiangsu Academy of Environmental
Science (JAES). There is a waterproof layer on the rooftop of the building but there was no guarantee that it was
made of asphalt. Furthermore, despite this waterproof layer on the rooftop of the building, the interferences of
emissions from this layer were believed to be insignificant because: 1) The waterproof layer was covered by the
layer of concrete, which was further covered with a layer of ceramic tile; 2) The building had been built for three
years before the sampling campaign was started; 3) It was documented that the VOC emitted from asphalt mainly
included benzene, toluene, ethylbenzene and xylene (Gardiner and Lange, 2005). However, the levels of benzene,
toluene, ethylbenzene, m/p-xylene and o-xylene were lower than those observed in other urban, industrial and
rural environments in different regions (section 3.1, Zhang et al., 2012; An et al., 2014 and 2015; Mo et al., 2015,
2017; He et al., 2019). 4) The sampling inlet was about 2-3 m above the rooftop of the building. It should be
noted that there is a waterproof layer on the rooftop of the building. However, it is not sure that the waterproof
layer was made of asphalt. Furthermore, though there is a waterproof layer on the rooftop of the building, the
interferences of emissions from the layer were believed to be insignificant because: 1) The waterproof layer was
covered by the layer of concrete, which was further covered with a layer of ceramic tile; 2) The building has
been built for at least three years when the sampling campaign was started; 3) It was documented that the VOC
emitted from asphalt mainly included benzene, toluene, ethylbenzene and xylene (Gardiner and Lange, 2005).
However, the levels of benzene, toluene, ethylbenzene, *m/p*-xylene and o-xylene were lower than those observed
in other urban and industrial and rural environments in different regions  (Zhang et al., 2012; An et al., 2014 and
2015; Mo et al., 2015, 2017; He et al., 2019) (details in section 3.1). 4) The sampling inlet were about 2-3 m
above the rooftop of the building. The station is located in an urban area of Nanjing, and is surrounded by heavy
road traffic, residential buildings, a plant and flower market, and several auto repair shops (Figure 1). Nanjing,
located in the western part of the YRD region, is one of the most urbanized and industrialized areas in the world
and consequently experiences severe air pollution. The site is located downwind of both Nanjing city center and
the wider YRD region (Zhao et al., 2017; Zhou et al., 2017), and is therefore ideally placed to determine the
combined impacts of VOCs from both local and regional atmospheric pollution.

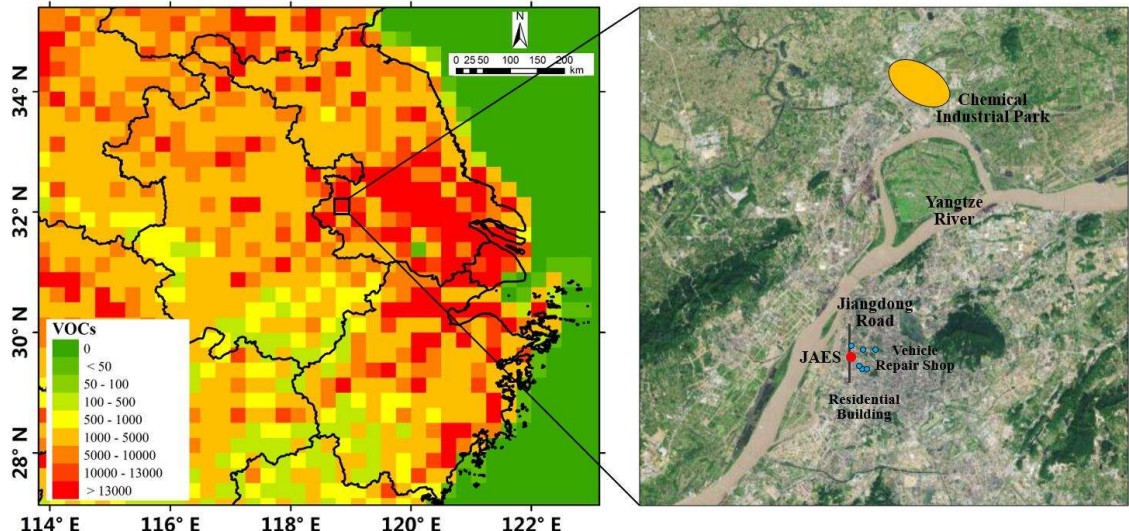

**Figure 1. (a) Maps of the study location showing VOCs emission at a resolution of 0.25 degrees (MG/a) (The data were from**
**MEIC emission inventory (www.meicmodel.org, last access date:15 September 2019). (b) The location of the JAES sampling**
**site is indicated by a red circle (The base map was from © Baidu Maps). The blue circles indicate vehicle repair shops, the**
**yellow circle indicates chemical industry park and the black solid line indicates a heavy traffic road**
Fifty-six VOC species including alkanes, alkenes, aromatics, and acetylene were measured at 1-h intervals using
a PerkinElmer Online Ozone Precursor Analyzer based on a thermal desorption-GC (gas chromatography)
system. First, the dried air samples were collected by a thermal desorption instrument and subsequently pre-
concentrated onto a cold trap. The sampling flow was 15 mL/min. After 600 mL of air was sampled, the cold
trap was heated to resolve the compounds adsorbed on to it. By applying the Dean's Switch technology whereby
the technology that transfers the effluent from one column to another column with a different stationary phase,
the low- and high-volatile components were injected into the $Al_2O_3/Na_2SO_4$ PLOT column (50 m × 0.22 mm ×
1 μm) and the dimethyl siloxane column (50 m × 0.32 mm × 1 μm), respectively, and analyzed using a flame
ionization detector (FID). The temperature increased from 46 °C for 15 min to 170 °C at a rate of 5 °C/min, and
then to 200 °C at a rate of 15 °C/min. The samples were finally held at 200 °C for 6 min.

A calibration was performed daily for quality control. The calibration curves showed good linearity with a
correlation coefficient of 0.99. Seven analyses were performed repeatedly to test the precision of the 56 species.
Calibrant concentrations in the gas standard mixture (56 $C_2$-$C_{12}$ NMHCs, Linde Spectra Environment Gases, Inc,
USA) ranged from 20 to 49 ppbC. The relative standard deviations of most of the 56 species were <5%,
representing an error of <0.5 ppb.
On the other hand, trace gases including CO, NO-$NO_2$-$NO_x$, $SO_2$, and $O_3$ were measured at 1-min resolution
using the commercial instruments of TEI 48i, 42i, 43i and 49i (Thermo Electron Corporation). All these
instruments were zero checked daily, span calibrated weekly and multi-point calibrated monthly. Furthermore,
meteorological conditions, including the temperature, relative humidity, pressure, wind speed and direction were
monitored at 1-min resolution by a weather station (Vantage Pro TM & Vantage Pro 2 plus TM Weather Stations,
Davis Instruments).

### 2.2. The PMF model for VOC source identification

In this study, the US EPA PMF (version 4.1) model, which has been widely used to conduct source apportionment
of VOCs (Zhang et al., 2013; Mo et al., 2017; He et al., 2019 and references therein), was applied to the observed
VOC data to identify potential VOC sources. A detailed description of the PMF model is provided by Yuan et al.
(2009) and Ling et al. (2011). In brief, the PMF model is a receptor model, which can identify the sources and
contributions of given species without prior input of their source profiles. In this study, a total of 25 species were
selected as the input for the PMF model including species with high abundances as well as typical tracers of
emission sources. Species with high percentages of missing values (> 25%) were excluded (i.e., 1,3-butadiene,
cis/trans-2-pentene, dimethylpentane, and trimethylpentane). The total concentration of the 25 selected species
accounted for ~92% of the total measured VOC composition. Furthermore, we calculated the total reactivity of
the selected 25 species to be ~90% of the total measured VOCs through the analysis of maximum incremental
reactivity (MIR) (Shao et al., 2009a). The high abundance and total reactivity contributions suggests that the
selected 25 species were appropriate for the PMF model simulation.
The PMF model was tested using a variety of factor numbers, and the optimum source profiles and contributions
were determined based on the correlation between modelled and observed data, the comparison of modelled
profiles with the results from emission-based measurements, and previous studies involving PMF/other receptor
model simulations (i.e., HKEPD, 2015; Wang et al., 2014; An et al., 2014; Liu et al., 2008a). For example,
different solution with different factor numbers was explored and the source apportionment results from a five-
factor resolution that could sufficiently explain the observed levels of VOCs were selected (details in Section
3.3). Compared with five-factor solution, the four-factor solution derived two profiles that attributable to gasoline
and diesel vehicular exhaust, while most of the aromatic species in these sources and certain amounts of $C_3$-$C_4$
species from fuel evaporation were categorized under industrial emission. On the other hand, the six-factor
solution has split a factor with high presence of ethyne and certain amounts of ethane (30% in species total), $C_3$
species and benzene (~20% in species total), while some alkenes (18-80% in species total) were incorporated
into fuel evaporation. Furthermore, the performance of the five-factor solution was evaluated using various
checks and sensitivity tests. Suitable correlations between the observed concentrations and those of each species
predicted by the model were observed, with the correlation coefficients ($R^2$) ranging from 0.60 - 0.91, indicating
that the solution adequately reproduced the observed variations of each species. All the scale residuals were
within ± 3σ with normal distributions for all species (Baudic et al., 2016). Moreover, different numbers of start
seeds were tested during the simulation and no-multiple solutions were found. The ratio of Q(robust)/Q(true)
obtained was ~0.93, close to 1 as suggested by previous studies and the user guide manual (Paatero, 2000; Lau
et al., 2010; Ling et al., 2016). In addition, the results from bootstrapping analysis for the five-factor solution
with bootstrap random seed found that all the factors were mapped to a basic factor in all the 20 bootstrap runs,
while the uncertainties of each species from bootstrapping analysis were within the range of 1~20%. In this study,
different $F_{peak}$ values ranging from -5 to 5 was tested in the 5-factor solution for a more realistic profile (Lau et
al., 2010; Baudic et al., 2016). The profiles with the nonzero $F_{peak}$ values were consistent with those with zero
$F_{peak}$ value, reflecting that there was little rotation for the selected solution, confirming that the profiles were
reasonably explained by the five-factor solution (Baudic et al., 2016). The results of $F_{peak}$ value = 0.5 (the base
run) was selected for analysis in this study. Overall, the above features demonstrated that the five-factor solution
from PMF could provide reasonable and stable apportionment results for the observed VOCs at the JAES site.
**3. Results and discussion**
**3.1 VOC observation statistics**
Table 1 shows the average concentration and standard deviation of fifty-six VOC species concentrations
measured at the JAES site, while Figure S1 in the supplementary presented the time series of all pollution data
collected at the JAES site. The annual average total VOC (TVOC, sum of the measured VOCs) concentrations
in 2016 was 25.7 ± 19.1 ppbv, with highest contributions from alkanes (13.6 ± 10.5 ppbv, ~53%), followed by
aromatics (4.4 ± 4.0 ppbv, ~17%), acetylene (4.5 ± 5.5 ppbv, ~17%) and alkenes (3.2 ± 3.3 ppbv, ~13%).
Annually, the most abundant 10 species were acetylene, propane, ethane, ethylene, butane, toluene, $i$-pentane, $i$-
butane, propylene and benzene, with a combined contribution of ~77% of the TVOC. This observed VOC
composition suggests that VOCs at the JAES site are predominantly sourced from combustion emissions (i.e.,
vehicular emissions). Alkenes are mainly associated with vehicular emissions and are more photochemically
reactive relative to alkanes and aromatics. The alkenes were found to have higher mixing ratios during weekdays
relative to the weekends (3.5 ± 0.2 vs 2.9 ± 0.1 ppbv for weekdays and weekend, respectively, $p < 0.05$), further
confirming the dominant contribution of vehicular emissions to VOC levels at the JAES site.

**Table 1. The average mixing ratios and standard deviation of VOC species concentrations measured at the JAES site from**
**January to December 2016.**

| Species | Average ± Standard deviation (ppbv) | Species | Average ± Standard deviation (ppbv) |
|---|---|---|---|
| **Alkanes** | **13.64 ± 10.53** | **Alkenes** | **3.24 ± 3.28** |
| ethane | 3.63 ± 2.68 | ethene | 1.72 ± 2.00 |
| propane | 3.70 ± 3.01 | propylene | 0.92 ± 1.16 |
| $i$-butane | 1.03 ± 0.87 | 1-butene | 0.12 ± 0.16 |
| $n$-butane | 1.55 ± 1.26 | $cis$-2-butene | 0.06 ± 0.09 |
| cyclopentane | 0.08 ± 0.10 | $trans$-2-butene | 0.16 ± 0.11 |
| $i$-pentane | 1.15 ± 1.24 | 1-pentene | 0.03 ± 0.03 |
| $n$-pentane | 0.61 ± 0.60 | $cis$-1-pentene | 0.02 ± 0.03 |

| | | | |
|---|---|---|---|
| 2,2-dimethylbutane | 0.02 ± 0.02 | *trans*-2-pentene | 0.02 ± 0.03 |
| 2,3-dimethylbutane | 0.05 ± 0.07 | isoprene | 0.14 ± 0.20 |
| 2-methylpentane | 0.26 ± 0.29 | *n*-hexene | 0.05 ± 0.03 |
| 3-methylpentane | 0.16 ± 0.21 | **Aromatics** | **4.40 ± 4.01** |
| *n*-hexane | 0.40 ± 0.45 | benzene | 0.80 ± 0.70 |
| methylcyclopentane | 0.26 ± 0.27 | toluene | 1.40 ± 1.35 |
| cyclohexane | 0.10 ± 0.16 | ethylbenzene | 0.50 ± 0.62 |
| 2,4-dimethylpentane | 0.03 ± 0.01 | *m/p*-xylene | 0.70 ± 0.71 |
| 2,3-dimethylpentane | 0.03 ± 0.02 | *o*-xylene | 0.25 ± 0.24 |
| 2-methyhexane | 0.06 ± 0.09 | styrene | 0.12 ± 0.17 |
| 3-methylhexane | 0.07 ± 0.10 | *n*-propylbenzene | 0.03 ± 0.03 |
| heptane | 0.09 ± 0.11 | *i*-propylbenzene | 0.03 ± 0.04 |
| methylcyclohexane | 0.07 ± 0.09 | *m*-ethyltoluene | 0.11 ± 0.14 |
| 2,2,4-trimethylpentane | 0.02 ± 0.03 | *p*-ethyltoluene | 0.05 ± 0.07 |
| 2,3,4-trimethylpentane | 0.02 ± 0.01 | *o*-ethyltoluene | 0.04 ± 0.05 |
| 2-methylheptane | 0.02 ± 0.02 | 1,3,5-trimethylbenzene | 0.04 ± 0.06 |
| 3-methylheptane | 0.02 ± 0.02 | 1,2,4-trimethylbenzene | 0.15 ± 0.21 |
| octane | 0.04 ± 0.06 | 1,2,3-trimethylpentane | 0.10 ± 0.14 |
| nonane | 0.02 ± 0.02 | *m*-diethylbenzene | 0.03 ± 0.06 |
| decane | 0.04 ± 0.04 | *p*-diethylbenzene | 0.04 ± 0.08 |
| undecane | 0.04 ± 0.07 | **Acetylene** | **4.47 ± 5.49** |
| dodecane | 0.09 ± 0.20 | -- | -- |

The TVOC level in this study was lower than previous measurements from an industrial site in Nanjing, in which 43.5 ppbv TVOC was reported (An et al., 2014). However, the high TVOC levels are likely due to the proximity of the observation site (~3 km northeast) to the Nanjing chemical industry area, as well as several iron, steel, and cogeneration power plants (within 2 km) (An et al., 2014). The variability in land-use between these two studies have also resulted in distinct VOC component profiles. In the industrial area, the relative contributions of alkenes and aromatics were as high as 25% and 22%, while the contribution of alkynes was only 7% (An et al., 2014). The alkane, alkene, and aromatic concentrations from the industrial site were 1.4, 3.4, and 2.2 times higher than the concentrations of this study, respectively, while alkyne concentrations were ~30% lower. Given the large variability observed between the two sites, it is crucial to assess the spatial variability of ambient VOCs across the city through a collaboration of multiple research groups using available real-time and online VOC monitoring systems.

Table S1 compares reported ambient VOCs from continuous measurements of ≥1 year in several megacities in a number of countries, including China. Continuous online measurements of ambient VOCs have only been available in China since 2010, unlike many developed countries whereby online VOC measurements have been available for multiple decades. In China, such measurements are only concentrated in a few megacities, including Beijing, Guangzhou, and Shanghai. The TVOC level reported in Nanjing was close to levels measured in

Shanghai (another megacity in the YRD, East China, 27.8 ppbv) (Wang et al., 2013), Tianjin (a megacity in
North China, 28.7 ppbv) (Liu et al., 2016), and Wuhan (a megacity located in central China, 24.3 ppbv) (Lyu et
al., 2016), but was considerably lower than Beijing (north China, 35.2 ppbv) (Zhang et al., 2017) and Guangzhou
(south China, 42.7 ppbv) (Zou et al., 2015). Alkanes were the dominant hydrocarbon group in all the cities;
however, some differences in relative contributions of the four classes were observed. The contribution from
aromatics was highest in Shanghai (31%) relative to the other cities, which is likely explained by the large
petrochemical and steel industry in Shanghai (Huang et al., 2011; Wang et al., 2013). In comparison, the
contribution of aromatics in Guangzhou (Zou et al., 2015) and the industrial area in Nanjing (An et al., 2014)
were 24% and 22%, respectively, while in other cities the contribution ranged between 17-19%. The current
ambient VOC concentrations in Chinese megacities are generally comparable to the urban VOC levels in
developed countries during the year 2000. However, in developed countries, the mixing ratios of VOCs were
observed to decrease in the recent decades following the implementation and formulation of VOC strategies
(Warneke et al., 2012). For example, the mixing ratios of VOCs in Los Angeles have decreased significantly
from 1960-2002 at an average annual rate of ~7.5%, while the mixing ratios of VOCs in London presented a
higher and faster decreased since 1998 when there were higher VOC mixing ratios than those in Los Angeles,
confirming that the earlier implementation of VOC reduction strategies in California had clearly led to the earlier
improvement of air quality compared to London (Warneke et al., 2012; von Schneidemesser et al., 2010).
Chinese megacities are therefore experiencing significnatly higher ambient VOCs contamination, given the
remarkable decrease in VOC emissions in developed countries over the last two decades (Pan et al., 2015;
European Environment Agency, 2016; U.S. EPA, 2017;). High VOC levels in Chinese megacities are known to
impact ambient ozone and secondary particle pollution, as well as cause adverse impacts on human health.
However, as China has a solid foundation for VOCs monitoring and control, numerous strict, appropriate and
targeted reduction strategies for VOCs have been/are being formulated and implemented in Chinese megacities
(Guo et al., 2017). It is expected these measures could help China to reduce VOC emissions/mixing ratios and
improve air quality in the future.

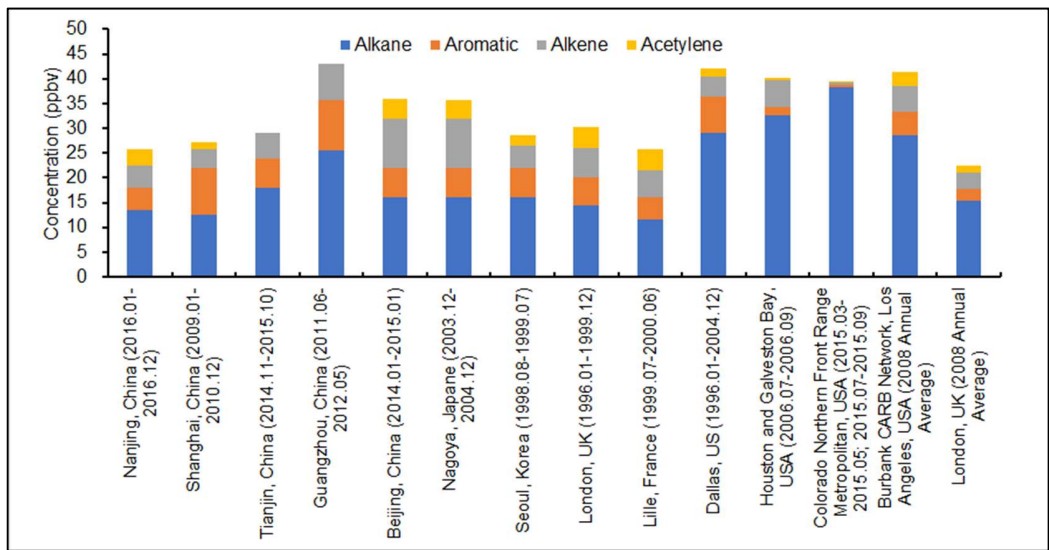

**Figure 2. Comparison of annual average concentrations of ambient VOC in different cities based on real-time online continuous measurements of at least one year.**

**3.2 Temporal variability**

In this study, ambient VOCs showed significant seasonal variability, with relatively high monthly average concentrations in winter (40.2 ± 24.0 ppbv) and spring (23.8 ± 15.0 ppbv), and low concentrations in summer (18.5 ± 14.6 ppbv) and autumn (20.1 ± 12.2 ppbv). As shown in Figure S2, the highest monthly average concentration was observed in December, followed by January. High pollution levels during the winter period are usually expected and is explained by atmospheric temperature inversions caused by cooler weather, which inhibits particle dispersion. Lower concentrations during the summer period are due to both favorable diffusion conditions and photochemical degradation of VOCs.

High wintertime VOCs pollution were also reported in Shanghai (Wang et al., 2013), Guangzhou (Zou et al., 2015), and Tianjin (Liu et al., 2016), though some differences in the monthly VOC variability were also observed. Except for the winter months, similar (and relatively stable) ambient VOC levels in the remaining months were observed for Guangdong (Figure 3). In Shanghai, relative high levels of VOCs were observed from October to January of the following year and from June to July based on the two-year measurement conducted from 2009 to 2010 (Wang et al., 2013). The inversion layer, the effect of cold front or uniform pressure in winter resulted in high levels of VOCs from October to January of the following year, while the frontal inverted trough or frequently observed stagnant high pressure system with southwest flow that could lead to poor diffusion were unfavorable meteorological conditions for high VOC levels from June to July. In addition, air masses transported from upwind chemical and petrochemical industrial factories located in the southwest and south of the monitoring site was another factor for the high VOC levels in summer (*i.e.,* June and July) and winter. VOCs concentrations in Tianjin showed significant monthly variability. Highest concentrations were reported in autumn and lowest concentrations were reported in summer. The observed monthly variability is affected by

several factors including the type and level of emissions and local meteorological conditions.

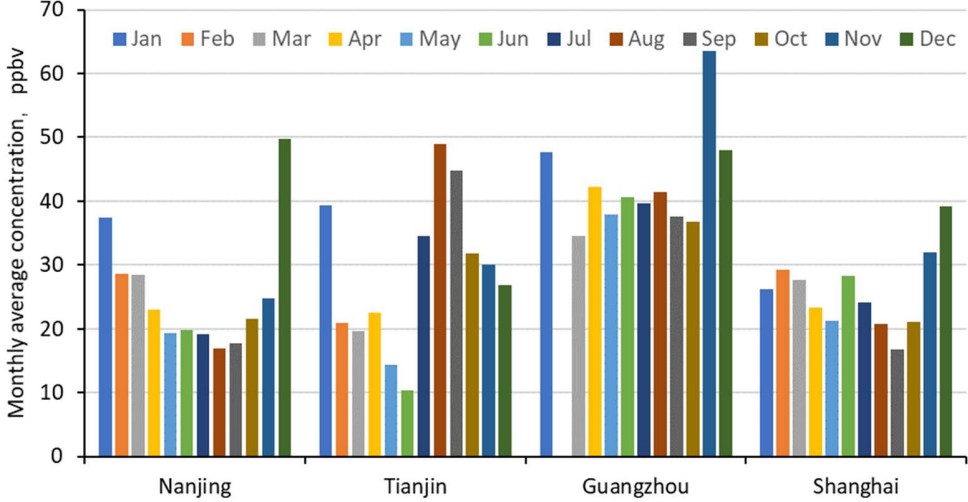

**Figure 3. Monthly variability of ambient VOCs at the JAES site and three other Chinese cities, Shanghai (Wang et al., 2013),**
**Guangzhou (Zou et al., 2015), and Tianjin (Liu et al., 2016).**
Figure S3 shows the diurnal trends in ambient VOCs for each month. The diurnal patterns were generally similar
for all the months. The observed peak at approximately 8-9 am (local time) corresponds with the city's morning
traffic rush. The concentration begins to decrease after 9 am, with lowest concentrations observed at
approximately 3 pm. The observed decline was likely due to reduced vehicle emissions, growth of the inversion
top, and enhanced photochemical VOC degradation. After 3 pm, the concentrations begin to increase gradually
as a result of increased vehicle emissions during the evening rush hour, as well as a reduction in the atmospheric
mixing height under evening meteorological conditions. The second evening VOC peak was less prominent than
the morning peak. Evening concentrations were generally higher than the daytime concentrations, and the
amplitudes of diurnal variability were larger in autumn and summer compared to winter and spring.

**3.3 Source apportionment of VOCs**
In this study, we applied the PMF model to apportion the sources of VOCs at the sampling site. Figure 4
illustrates the source profiles of the VOCs produced by the PMF model. Five VOC sources were resolved by
PMF, including biogenic emissions (Source 1), fuel evaporation (Source 2), gasoline vehicular exhausts (Source
3), diesel vehicular exhausts (Source 4), and industrial emissions (Source 5).

Source 1 was identified as biogenic emissions due to the high loading of isoprene – a typical tracer of biogenic
emissions (Lau et al., 2010; Yuan et al., 2012). Source 2 was represented by high proportions of 2-methylpentane,
3-methylpentane, *i*-pentane, and cyclopentane. Pentanes are mainly associated with profiles from gasoline-
related emissions (Barletta et al., 2005; Tsai et al., 2006). However, the low contributions of incomplete
combustion tracers in this profile suggested that the VOCs were sourced from fuel evaporation. The high

presence of pentanes in this profile was consistent with the source profile of gasoline volatilization extracted from principal component analysis/absolute principal component scores (PCA/APCs) based on the observed VOC data collected in an industrial area of Nanjing (An et al., 2014), the source profile of gasoline evaporation from PMF at the suburban site and urban site in Beijing and Hong Kong (Yuan et al., 2009; Lau et al., 2010). Particularly, based on the emission-based measurement, Liu et al. (2008b) conducted source apportionments of VOCs in the Pearl River Delta region by the chemical mass balance (CMB) receptor model, which attributed the source with high loadings of $n/i$-pentanes, cyclopentane and 2/3-methylpentane as gasoline evaporation. Therefore, Source 2 here was identified as fuel evaporation.

Source 3 and Source 4 were identified as vehicular exhausts due to their high loadings of incomplete combustion tracers, i.e., $C_2$-$C_4$ alkanes and alkenes (Guo et al., 2011a, b; Zhang et al., 2018). Zhang et al. (2018) compared the VOC composition of vehicular emissions from Zhujiang Tunnel in 2014 and 2004 in the Pearl River Delta region with those from other tunnel measurements. $C_2$-$C_4$ alkanes and alkenes were found to made the greatest contributions to the loading of VOCs emitted from vehicles in 2014. The higher proportions of $n/i$-pentane, $n$-hexane, and methylcyclopentane in Source 3 relative to Source 4 indicated VOCs sourced from gasoline vehicular exhausts (Liu et al., 2008b; Guo et al., 2011b; Zhang et al., 2018). Source 4 was identified as diesel vehicular exhausts due to the high percentages of ethyne, ethane, and propene, as well as $C_2$-$C_4$ alkenes (Ho et al., 2009; Cai et al., 2010; Ou et al., 2015; Liu et al., 2008c). Source 5 was characterized by high concentrations of aromatics. In addition to gasoline vehicle emissions, industrial emission could be another important contributor to ambient aromatic hydrocarbons in the Yangtze River Delta, Pearl River Delta and North China Plain (Yuan et al., 2009; Zhang et al., 2013, 2014; An et al., 2014; Mo et al., 2015, 2017; He et al., 2019). The tunnel studies and emission-based measurement results found that aromatic hydrocarbons from gasoline vehicle exhaust were coherently emitted with pentanes, butenes, $n$-hexane, and cyclopentane, which were more consistent with the profile in source 3 mentioned above (Liu et al., 2008; Ho et al., 2009; Yuan et al., 2009; Zhang et al., 2018). Therefore, the absence of above species in source 5 indicated that this source could be related to industrial emission (Zhang et al., 2014). Particularly, the high presence of toluene, ethylbenzene, xylenes, ethyltoluene and trimethylbenzene was consistent with the emission-base measurement results conducted in paint and printing industries (Yuan et al., 2010) and manufacturing facilities (Zheng et al., 2013). On the other hand, the profile of high presence of aromatic hydrocarbons ($C_7$-$C_9$ aromatics) and the certain amount of ethene, was also agree with the profiles measured in the areas dominated by industrial emissions in the Yangtze River Delta region (An et al., 2014; Shao et al., 2016; Mo et al., 2017). For example, An et al. (2014) reported that toluene, ethylbenzene, xylenes, and trimethylbenzenes could be emitted from different industrial processes, and identified that the factors with high loadings of these species as industrial production, solvent usage and industrial production volatilization sources by PAC/APCS at the industrial area in Nanjing. On the other hand, Mo et al. (2017) identified the factors with high concentrations of $C_7$-$C_9$ aromatics and ethene as residential solvent usage, chemical and paint industries and petrochemical industry with the PMF model applied to the data

collected in an industrialized coastal city of Yangtze River Delta region. To further identify source 3 and source 5, the ratio of toluene/benzene (T/B, ppbv/ppbv) in each profile was compared with those obtained from emission-based measurements and tunnel study results (Zhang et al., 2018 and references therein). The ratios of T/B were ~8.2 and ~1.2 for sources 5 and 3, respectively, and were consistent with those of "industrial processes and solvent application", and "roadside and tunnel study", respectively (Zhang et al., 2018 and references therein). This further confirmed that source 3 was related to gasoline vehicular exhaust, while source 5 was associated with industrial emission.

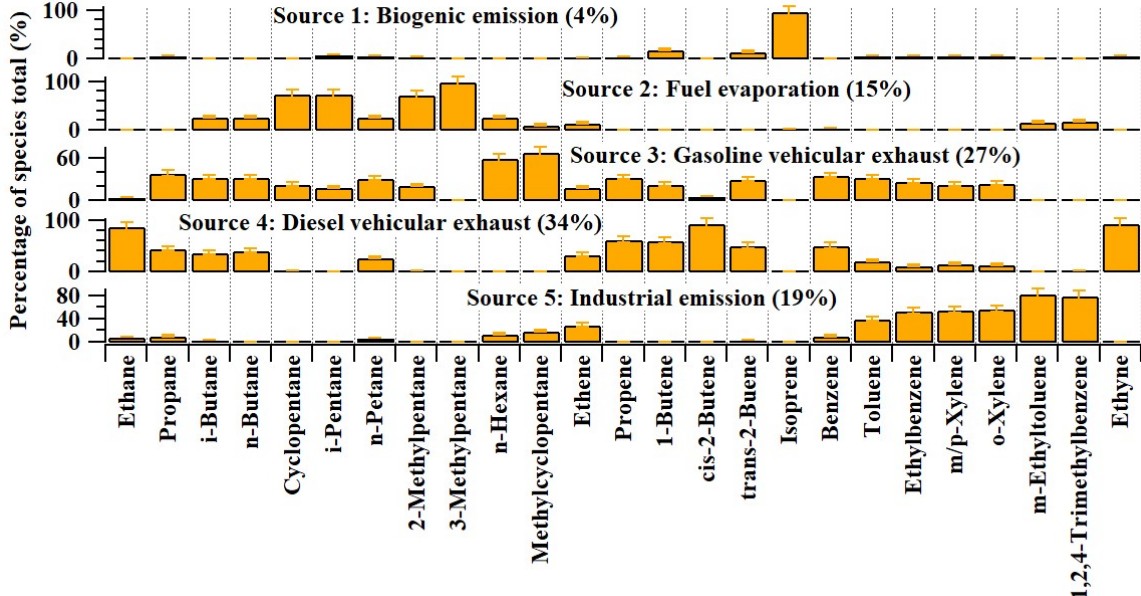

**Figure 4. Source profiles of VOCs identified using the PMF model and the relative contributions of the individual VOC species.**

Vehicular exhausts were found to be the most significant contributor to the TVOCs at the JAES site, with average contributions of ~34% and ~27% for diesel and gasoline exhausts, respectively, followed by industrial emissions (19%), fuel evaporation (~15%), and biogenic emissions (~4%). Our results are inconsistent with previous results observed at industrial sites in Nanjing (An et al., 2014; Xia et al., 2014a). An et al. (2014) found that industrial activities were the most significant source of VOCs, contributing 45%-63% (mainly aromatic VOCs), followed by vehicle emission at 34%-50%. Similarly, Xia et al. (2014a) reported solvent usage and other industrial sources to account for most (31%) of the VOCs in a suburban site in southwestern Nanjing, close in proximity to Nanjing's industrial zone. Fossil fuel/biomass/biofuel combustion were the second highest contributors at 28%, while the average contribution of vehicular emissions was 17%, mainly from the northern center of Nanjing (Xia et al., 2014a). Combined, these results infer vehicular emissions to be a major component of urban emissions in Nanjing. The observed spatial variability in the contributions of VOC sources infers the complex emissions characteristics of VOCs in Nanjing, likely due to the city's unique industrial structure. For example, the sampling site (i.e., the JAES site) was located at a more residential and urban area compared to other sites listed in An et

al. (2014) and Xia et al. (2014). There are more than 0.22 million people living in the areas surrounding the sampled site (within 3 km of the observation site) which composed of residential communities, schools, government agencies, and business centers. These results also demonstrate that local emissions are dominant contributors to ambient VOCs levels in Nanjing.

The dominant contribution of vehicular emissions to ambient VOCs in Nanjing is consistent with the urban/central areas of other large cities, including Hong Kong, Guangzhou, Shanghai, and Beijing, as identified and quantified by the PMF model (Yuan et al., 2009; Cai et al., 2010; Guo et al., 2011a; Zhang et al., 2013; Wang et al., 2015). In addition, our results are in agreement with the anthropogenic VOC source emission inventory of Jiangsu Province in 2010 (Xia et al., 2014b), indicating vehicular emissions and industrial emissions (i.e., solvent usage and industrial process source) to be the two dominant sources of VOCs in the region. However, the contributions of vehicle related emissions (i.e., ~25%) and industrial emissions were lower and higher than those quantified by the PMF model in this study, respectively. The observed discrepancy between the two studies may be due to differences in source categories, measured VOC species, and/or sampling locations and methods used in the different models. For example, the VOC sources in Jiangsu province were categorized into vehicular related emission (~26%), industrial solvent usage (~25%), fossil fuel combustion (~24%), industrial processes (~22%) and biomass burning (~3%). Further, vehicle related emissions only included emissions from motor vehicles and ships, and the volatilization of fuel, while solvent usage included organic solvents volatilized from a variety of industries (the industrial produce process of electronic equipment manufacturing, furniture manufacturing, printing, packaging, inks, adhesives, etc. and other dry cleaning, catering, and architectural decoration processes). Higher vehicular emission contribution in this study may also be due to the increasing number of vehicles from 2010-2014 as a result of increased urbanization and industrialization (Statistical yearbook of Nanjing, 2014).

Figure 5 illustrates the mean diurnal variability of all identified source at the JAES site. These trends were influenced by the variability in emission strength, mixing height, and the concentrations and photochemical reactivity of individual species in each source profile. For example, we observed a typical diurnal pattern with a broad peak between 9 am-6 pm for biogenic emissions, as the emission rate of isoprene from vegetation is largely depended on ambient temperature and sunlight intensity. Higher levels of diesel and gasoline vehicular emissions were observed in the evening and early morning due to a reduced mixing height and increased emissions from the morning and evening rush hour. Lower concentrations observed during daytime hours were likely due to decreased emissions, an increased mixing height and enhanced photochemical loss (Gillman et al., 2009; Yuan et al., 2009; Wang et al., 2013). A diurnal pattern of fuel evaporation that was similar to that of vehicular emissions. Though the evaporation of fuel is dependent on temperature, the average temperature in the morning and evening (*i.e,* 0800-1000 and 1700-1900 LT, respectively) when peaks of fuel evaporation were found was only about ~1.2 °C lower than that observed from noon to afternoon (1100-1600 LT), which may not result in

much higher fuel evaporation at noon (the difference between maximum and minimum values for fuel
evaporation was found to be ~6 μg/m³). On the other hand, in addition to evaporation from the gas station, fuel
could evaporate from hot engines, fuel tanks and the exhaust system when the car is running. Furthermore, the
engine remains hot for a period of time after the car is turned off, and gasoline evaporation continues when the
car is parked (Technology center, University of Illionois, https://mste.illinois.edu/tcd/ecology/fuelevap.html,
access date: 25 December 2019). The similarity of diurnal variations of fuel evaporation to vehicular emissions
suggested that the prominent peak in the morning and evening hours were related to the increased vehicles in
the traffic rush hour and emissions accumulated in the relatively low boundary layer. Moreover, we identified
higher concentrations of industrial emissions at night and in the early morning, with values remaining fairly
stable during daytime hours. This finding is consistent with other observations in urban and rural areas (Yuan et
al., 2009; Leuchner and Rappenglück, 2010).

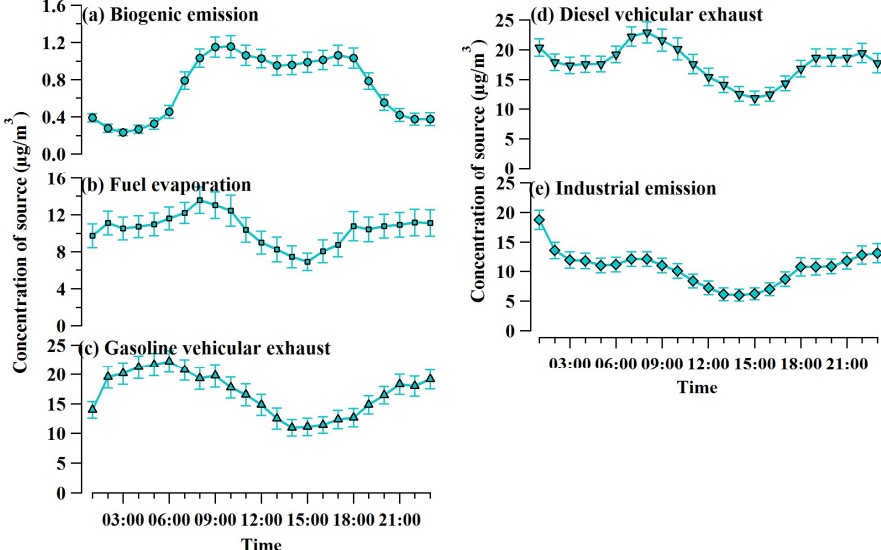

**Figure 5. Diurnal patterns in source concentrations of the five identified sources**
**3.4 Contributions of VOC sources to O₃ formation**
As important $O_3$ precursors, information on the contributions of VOC sources and related species to $O_3$ formation
is necessary for the formulation and implementation of VOC control measures. To achieve this goal, the
Maximum Incremental Reactivity (MIR) method, which evaluates the $O_3$ formation potential (OFP) on the basis
of mass concentrations and maximum incremental reactivities of VOCs with the OH radical, were adopted in
the present study (Shao et al., 2009b, 2011; Mo et al., 2017). Figure 6 presented the relative contributions of
individual VOC sources and related VOC species from PMF to OFP at the JAES site. Industrial emissions were
found to have the largest OFP at JAES due to the high loadings of aromatic VOC species that have relatively
high OH reactivities in this source profile (Atkinson and Arey, 2003), with the OFP value of ~43 μg/m³ and the
contribution percentage of ~32% to the total OFP of all VOC sources, followed by diesel vehicular exhausts
(~36 μg/m³, ~27%), gasoline vehicular exhausts (~32 μg/m³, ~24%) and fuel evaporation (~13 μg/m³, ~10%).
Furthermore, though the MIR value of isoprene was much higher than other VOC species, biogenic emissions
only contributed ~7% (~9 μg/m³) to the total OFP of all VOC sources as the relatively low mixing ratio of
isoprene at the JAES site. Similarly, using the same method to evaluate OFP of different VOC sources, Mo et al.
(2017) found that industrial emissions (including the emissions of petrochemical industry, chemical and paint
industries, solvent usage) and vehicular emissions were the dominant VOC sources of the total OFP in an
industrialized coastal city (i.e., Ningbo) in the YRD region. Therefore, our results further demonstrated the need
to minimize VOC emissions from industrial emissions and vehicle exhausts in order to lower O₃ formation and
photochemical pollution in YRD.

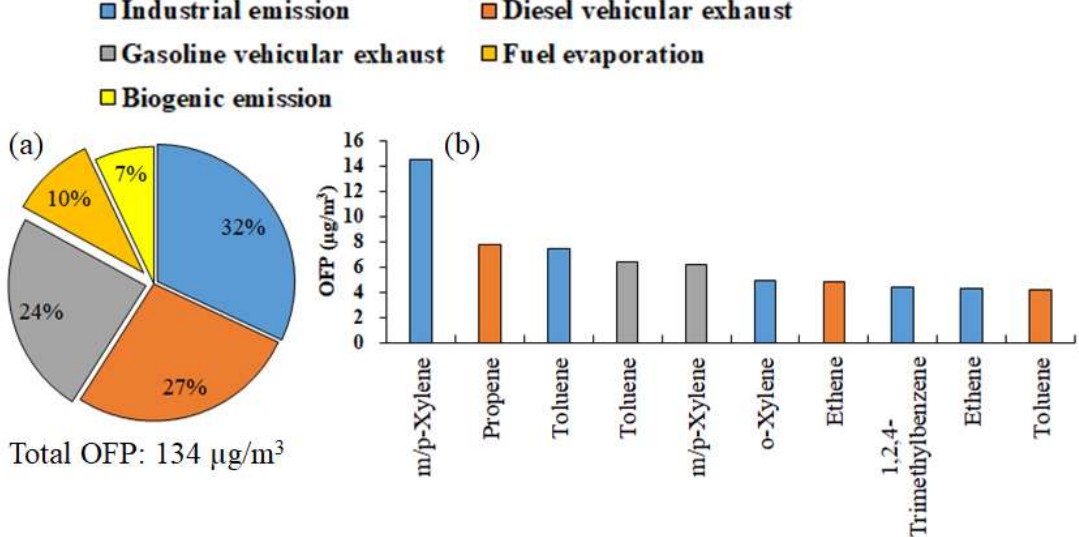


**Figure 6. (a) The contribution of individual sources to the total OFP of all sources extracted from PMF and (b) OFP values**
**of the top 10 VOC species in the different source categories.**

Based on the mass concentrations of individual species in each source, we found that *m,p*-xylene and toluene in
industrial emissions and gasoline vehicular emissions, propene, ethene, toluene and *m,p*-xylene in diesel
vehicular emissions, and *o*-xylene, 1,2,4-trimethylbenzene and ethene in industrial emissions to be the dominant
species from VOC emissions contributing to photochemical O₃ formation. Thus, only a small number of VOC
species can be monitored for the effective control of O₃ formation.

**3.5 Policy summary and implications**
To effectively control photochemical pollution, the Prevention and Control of Atmospheric Pollution Act was
passed in 1987 and amended in 2015. As a result, a series of measures to prevent and control VOCs levels have
been and are being implemented by central and local governments, including the implementation of new laws
and regulations, and the advancement of technology. The results of this study suggest that photochemical O₃
formation within the urban areas of Nanjing city are VOC-limited, which is consistent with observations in the
urban locations of other regions, including the North China Plain, the Yangtze River Delta and the Pearl River
Delta. Minimizing VOC emissions and their concentrations should therefore be prioritized in order to alleviate
$O_3$ pollution in urban environments. The prevention and control of VOC pollution has been listed as one of the
key tasks of "the Blue Sky" Project initiated in 2012 by the Department of Environmental Protection of Jiangsu
Province. Furthermore, the administrative measures on the Prevention and Control of Volatile Organic
Compounds Pollution in Jiangsu (Order No. 119 of the Provincial Government) was enacted on March 6, 2018
and implemented on May 1, 2018, with the aim of controlling VOC emissions in Jiangsu Province.
In order to achieve these goals, various measures have been implemented (Table S2), including: 1) investigating
the current pollution status and identifying the progress of VOC prevention and control in Jiangsu Province
(Provincial Office of the Joint Conference on the prevention and control of air pollution [2012] No. 2); 2)
conducting a strict industry access system, under the Advice on Promoting Air Pollution Joint Prevention and
Control Work to Improve Regional Air Quality (Office of the State Council [2010] No. 33); 3) strengthening the
remediation on existing sources of VOCs and reducing VOC emissions from these sources, under the Guidelines
for the Implementation of Leak Detection and Repair (LDAR) in Jiangsu Province (Trial) (Provincial Office of
Environmental Protection [2013] No. 318); 4) strengthening the VOC monitoring capacity, under the Guidelines
for Control of Volatile Organic Compounds Pollution in Key Industries in Jiangsu Province (Provincial Office
of Environmental Protection [2013] No. 128); 5) improving standards regarding VOC emissions for key
industries, including standards for surface coating of the automobile manufacturing industry (DB32/2862-2016),
the chemical industry (DB32/3151-2016), and furniture manufacturing operations (DB32/3152-2016), which are
still effective since their enforcement; 6) implementing the Pilot Measures for Volatile Organic Compounds
Discharge Charges (Ministry of Finance [2015] No. 71) on October 1, 2015 to raise awareness pertaining to
emissions reduction in factories and to control VOC emissions from industrial sources; 7) encouraging the public
to live a low-carbon life and supervise and offer recommendations in accordance with the laws, under the
Measures for Public Participation in Environmental Protection in Jiangsu Province (Trial) (Provincial Regulation
of Environmental Protection Office [2016] No. 1).
Based on the VOC source apportionment results in this study, we identified vehicular emissions and industrial
emissions as the two major VOC sources contributing to photochemical $O_3$ formation. Other measures and/or
regulations have been conducted in the Jiangsu Province to effectively control VOC emissions from vehicles
and industry. For vehicular emissions, the Regulations on Prevention and Control of Vehicle Exhaust Pollution
in Nanjing was amended in July 2017, and subsequently in March, 2018
(http://hbt.jiangsu.gov.cn/col/col1590/index.html). The new regulation not only focusses on vehicle emissions,
but also incorporates a number of additional topics, including optimizing the function and distribution of urban
areas, limiting the number of vehicles in the region, promoting new green energy vehicles, and improving the
quality of fuel. The promotion of intelligent traffic management, implementation of a priority strategy for public
transportation, and construction of more efficient traffic systems to promote pedestrian and bicycle use is
recommended. Further studies should be conducted to estimate and manage the increasing quantity of vehicles
on the road. As of January 1, 2017, regulation stipulate that all new and used vehicles should meet the fifth phase
of vehicle emission standards, including vehicle manufacture, sales, registration and importation. For vehicles
already in use, an environmental protection examination should be conducted annually, based on the standards
of GB 14622-2016, GB 18176-2016, GB 19755-2016, and HJ 689-2014. Penalties are issued if qualified
vehicles excessively emit pollutants due to poor maintenance.

For industrial emissions, various policies have been implemented to reduce VOC emissions, particularly in
chemical industries: including, 1) investigations on the VOC emissions of the chemical industry and the
establishment of an archive system for VOC pollution control, particularly the inspection of industry information,
products and materials, unorganized emission of storage and exhaust gas treatment facilities, under the Plan for
Investigation of Volatile Organic Pollutant Emissions in Jiangsu Province, mentioned in the Provincial Office of
Environmental Protection [2012] No. 183; 2) exhaust gas remediation in the chemical industry park, under the
Technical Specifications for Prevention and Control of Air Pollution in Chemical Industries in Jiangsu Province
(Provincial Office of Environmental Protection [2014] No. 3), which requires the establishment of the long-term
supervision of exhaust gas remediation in the chemical industry park of Jiangsu Province; 3) a pilot project on
the leak detection and repair (LDAR) technology in the chemical industry park, under the notification on carrying
out the technical demonstration and pilot work of leak detection and repair (LDAR) in petrochemical and
chemical industries (Provincial Office of Environmental Protection [2015] No. 157). The TVOC removal
efficiency of organic exhaust vents should be >95%, and higher for areas of excessive environmental pollution
at >97% (GB 31571-2015).
Furthermore, though measures have been adopted to improve standards and control vehicle VOC emissions,
most of these policies only focus on total VOC emissions (or the mass of total emissions) and do not consider
the impacts of individual VOC species. To accelerate the implementation of existing policies and to strengthen
collaborative regional prevention and control, priority should be placed on specific high-impact VOC species
(i.e., *m,p*-xylene and toluene in the industrial emission and gasoline vehicular emission) by considering both
their reactivity and abundance.

Last but not the least, $O_3$ pollution is a regional cross-boundary environmental issue rather than a local pollution
problem. Apart from VOCs, $NO_x$ was another important precursor for $O_3$ formation with its dual roles in $O_3$
production (enhancing $O_3$ formation in non $NO_x$-saturated environment and titrating $O_3$ in $NO_x$-saturated
environment). In other areas (i.e., the rural environment and/or the downwind areas of urban center in the same
region) where the concentrations of $NO_x$ are low and/or there is a non $NO_x$-saturated environment, the situation
may be different and controlling VOCs should be conducted cautiously (Zheng et al., 2010; Yuan et al., 2013;
Ou, et al., 2016). Therefore, from a regional perspective, the benefits of VOCs control measures could be further
evaluated with those of $NO_x$ (i.e., the appropriate ratios of $VOC/NO_x$ for the reduction of $O_3$ pollution) as well
as the associated $O_3$-VOCs-$NO_x$ sensitivity. Therefore, one important concern for the policy formulation and
implementation system is whether controlling VOCs and $NO_x$ individually or controlling both VOCs and $NO_x$
is more effective and appropriate for alleviating $O_3$ pollution. It is necessary to consider the reduction ratios of
$VOC/NO_x$ when VOCs and $NO_x$ are simultaneously controlled. Finally, long-term monitoring studies are
necessary to determine the cost-benefits and performance of each policy.
**4. Conclusion**
In this study, a one-year field sampling campaign was conducted to investigate the VOC characteristics at an
urban site in Nanjing (the JAES site), Jiangsu province. In total, 56 VOCs including 29 alkanes, 10 alkenes, 16
aromatics and acetylene were identified and quantified. The composition analysis found that alkanes were the
dominant group of VOCs observed at the JAES site (~53%), followed by aromatics, acetylene, and alkenes. This
finding is consistent with the VOC measurements in studies conducted in the North China Plain, Pearl River
Delta, and Yangtze River Delta. We observed distinct seasonal patterns of TVOCs, with maximum values in
winter and minimum values in summer. Similarly, prominent morning and evening peaks were observed in the
diurnal variability of TVOCs, influenced by local emissions and meteorology.

Based on the observed VOC data, we identified five dominant VOC sources at the JAES site using a PMF model.
By considering both the abundance and maximum incremental reactivity of individual VOC species in each
source, the OFP values identified industrial and vehicular emissions, particularly *m,p*-xyleme, toluene and
propene, as the main contributors of $O_3$ pollution. Local governments have strengthened several measures to
minimize VOC pollution from vehicle and industrial emissions in the Jiangsu province in recent years, though
most of these policies focus particularly on lowering the total emissions of VOCs. Furthermore, from a regional
perspective, it is suggested that appropriate ratios of $VOC/NO_x$, their associated sensitivity to $O_3$ formation and
relative benefits/disbenefits of reducing $VOCs/NO_x$ should be investigated and evaluated when control measures
of VOCs and $NO_x$ were both conducted.

**Author Contributions.** Jun Bi, Zhenhao Ling, and Qiuyue Zhao designed the research and carried them out. Zhenhao Ling performed the data simulation. Qiuyue Zhao and Guofeng Shen performed the observation data analysis. Qiuyue Zhao prepared the manuscript with contributions from all co-authors.

**Competing Interests.** The authors declare that they have no conflict of interest.

**Acknowledgements.** This work was supported by National Key R&D Program of China (No.2016YFC0207607, 2017YFC0210106), National Science Foundation of Jiangsu Province of China (General Program, No. BK20161601), Open Research Fund of Jiangsu Province Key Laboratory of Environmental Engineering (No.ZX2016002) and National Natural Science Foundation of China (No. 41775114). This work was also partly supported by the Pearl River Science and Technology Nova Program of Guangzhou (grant no. 201806010146).

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
