# Peer review of "Sources of volatile organic compounds and policy implications for"

_Atmospheric Chemistry and Physics, 2019_

## Referee Comment (RC1) · Anonymous Referee #2 · 16 Nov 2019

In this study, the authors conducted one-year VOC observation at an urban site in Nanjing. They analyzed the seasonal and diurnal characteristics of 56 VOCs as well as their sources using the PMF model. A box model together with a Master Chemical Mechanism (MCM) was used to identify the relationships between the contributions of VOC sources and the O3 formation. The results were also compared with VOCs data from other Chinese megacities like Beijing, Guangzhou, and Shanghai. VOC have been well recognized to be responsible for the swift development of air pollution events since volatile organic compounds (VOCs) are key precursors of O3 and secondary

organic aerosols (SOA). However, the speciation and emission strength of these VOC have been demonstrated to be hard to acquire due to the fact that VOC can be emitted from a diversity of domestic and industrial activities. Therefore, fiżeld measurements of VOC emissions are critically needed in China. This work can be a signifiżcant contribution to the atmospheric research community. Overall, the manuscript is fairly well written and I would recommend the manuscript for publication after minor revisions.

Specific concerns: 1. Line 61: what's "photochemical industry"? 2. Sample location: there is always an asphalt waterproof layer on the rooftop of an office building. How to avoid the interference of this emission? 3. Aromatics are important of gasoline. So, source 5 could be also identified as gasoline cars. Thus, the identification of source 5 to industrious emissions maybe need more relevant tracers. 4. Why the diurnal trend of fuel evaporation showed a decrease at noon time since this source is temperature dependent?
* * *

---

## Referee Comment (RC2) · Anonymous Referee #1 · 19 Dec 2019

Zhao et al. describe VOC measurements conducted at the Jiangsu Academy of Environmental Science (JAES) in Nanjin, China. The authors measure VOCs using a GC system, and interpret the sources of these VOCs using positive matrix factorization. The authors evaluate the environmental impacts of these emissions on ozone formation using an observation-based model (OBM) employing the Master Chemical Mechanism (MCM v3.2), and identify the anthropogenic VOCs likely to be significant ozone precursors. The authors also evaluate ozone sensitivities to VOC and NOx reductions, and conclude that VOC reductions would be the best strategy to reduce ozone in Nanjing.

In general, the manuscript reads very well and is well-organized to tell a coherent message. I appreciate the authors work to carefully measure VOCs and benchmark these measurements against other cities in China. I am generally convinced by the PMF results given that the authors interpretation is reasonable, and the PMF factors are prescribed to obvious sources in the Nanjing area (which are very well described); however, I do have some recommendations that could improve the PMF analysis and strengthen the justification of source apportionment. Finally, I believe the use of the OBM is justified to evaluate VOC RIR, but I am not convinced that the OBM can be used to evaluate the ozone isopleth without further evidence that the model is doing an adequate job to capture ozone formation in the Najning region. My comments below primarily address PMF and the OMB.

Major comments

1.  The PMF solution appears to be reasonable; however, I believe the authors need to do more to show that the PMF solution is robust. In Section 2.2, the authors state that comparisons were made to observations, emissions inventories, and previous PMF analyses, but no evidence is shown here or in the supplement to convince the reader that this is true. Can the authors show the Q/Qexp and explain why they settled on a 5-factor solution? What was the factor space used? Did the authors vary other parameters (e.g. Fpeak) or conduct a bootstrapping analysis to estimate uncertainty? Can the authors show the comparisons to other factor profiles reported in literature (e.g. the industrial factor compared to An et al. 2014)?

    I ask because PMF is partly subjective, and a more thorough discussion is necessary to justify why the authors settle on the solution presented in the manuscript. A 5-factor solution seems reasonable, and the factors discussed all appear to be consistent with the sources surrounding the sampling site, but this should be shown with more evidence in the main text or supplemental information.

2.  The authors employ an OBM to evaluate ozone sensitivity to VOCs and NOx. OBMs are primarily useful because they allow one to evaluate relative incremental reactivity (as the authors describe in Section 2.3). One strength of an OBM is that you do not need all of the measurements that describe ozone formation; rather, you calculate source functions that explain residual effects on the time evolution of a measured species (e.g. meteorology, chemistry not accounted for in the mechanism, additional precursors that contribute to

ozone formation, etc). From these calculations, you can derive the RIR by conducting a small perturbation on the system (e.g., decreasing or increasing the concentration of a species that is measured and well-represented by the model). The calculations of RIR are good and justified with the use of an OBM.

In Section 3.4, the authors extend this analysis to evaluate the ozone isopleth. In this context, I don't believe the use of an OBM is justified. Isopleth calculations are defendable if a large fraction of the local, photochemically produced ozone is explained by the measured precursors. If a significant fraction of this produced ozone is explained by the time-dependent source function (i.e., the "residual" ozone), then the authors may not be measuring (or including in the model) a significant fraction of the VOC precursors needed to derive ozone formation. In that case, how can the authors determine whether Nanjing is VOC or NOx-sensitive? The isopleth presented in Fig. 6 is very NOx saturated, which the authors say generally agrees with previous literature. But do the measurements really defend this? The argument that Nan

If the authors are to present an ozone isopleth, then I believe there needs to be a much larger discussion describing how well the OBM performs in reproducing observed ozone mixing ratios. Without much discussion, I can only assume that there is residual ozone that is explained by the time-dependent source functions derived through OBM calculations, and not by the precursors measured by GC. How much of the ozone calculated via OBM is explained by the precursors measured by the GC, and how much of the ozone is unexplained? Can the authors show an analysis (perhaps just a time series) showing ozone explained by the precursors, and ozone explained by the source function? This helps place into context the extent to which the measured VOCs were the primary contributors to ozone observed at the ground site.

Finally, the authors also need to provide more details about the OBM itself. The only description of how the model was tailored to the Nanjing observations is provided at lines 135-140. What meteorological conditions were used? If this is observation-based, I assume that dilution by PBL expansion and wind speed are lumped into the source functions, but what about incident solar radiation? How do the authors calculate photolysis frequencies? Did the authors use a model, such as TUV, or was there a solar spectrum measurement? Can the authors provide a jNO2 frequency to orient the reader? The authors constrain CO, NOx, SO2, and O3. How were these species measured, what instrumentation, and how was this instrumentation calibrated? Finally, when were the O3 episodes? A time series showing ozone over the course of the campaign would be helpful.

**Other comments**

Line 112: PMF can be conducted using many tools. Is this the US EPA model, SoFi, another model, or one that was developed by Yuan et al. or Ling et al.? This should be noted here, with a relevant reference if necessary.

Line 120: Shao et al. discuss VOC reactivity through analysis of maximum incremental reactivity (MIR) and by calculating propylene-equivalent concentration. Which method are you referring to?

Line 130: please provide references for the MCM (see the following website for appropriate references depending on the sub mechanisms used: http://mcm.leeds.ac.uk/MCMv3.2/citation.htt).

Also, is there a reason that v 3.2 was used, rather than 3.3.1? v3.3.1 has updates to the isoprene mechanism that may (or may not) be relevant here.

Line 164: By TVOC, you mean the sum of measured VOCs?

Line 172: Is this reversed? The first number (referring to weekdays) is lower than the second (referring to weekends).

Table 1: You only give an average and standard deviation - no mixing ratio ranges are shown. I recommend removing "range"

Line 194: Continuous VOC measurements have been available much longer than this in other countries. I would recommend changing this wording to say "online VOC measurements have been available for multiple decades"

Fig 2. This is a nice benchmark of the Nanjing measurements with other cities during a period when developed countries were still reducing mobile emissions (mid 1990s - early 2000s). How does this compare with measurements conducted in developed countries today? It would be nice to see how the mixture in Nanjing compares to London or Los Angeles today, and would also highlight the gap that could be achieved with further VOC reductions.

Lines 227-228: Can the authors briefly summarize the conclusions from Wang et al. 2013? Was it due to changes in prevailing winds, or simply due to a buildup of pollutants during strong inversions?

Line 280-281: As the authors note, these differences result, in part, due to the proximity of the different sampling campaigns. However, I think it's also good to note why these differences are important. How much of the population resides in the sampled region? Is the mix measured in a more residential area more important for human exposure? This is certainly a nice motivation to look at the spatial VOC distributions in Nanjing in the future.

Line 319: Do you mean that you averaged the PMF solutions during the ozone episodes and non-episode days to look at differences?

Figure 6. The isopleth description is somewhat confusing - is this % change in NOx and VOCs, or % of base-case VOCs?

Section 3.5. Without more work to convince the reader that the ozone isopleth is reasonable, I believe these statements would need to be amended. First, the authors haven't shown that the ozone precursors measured account for the majority of the ozone modeled in the OBM. Second, the recommendation to prioritize VOC reductions (line 381) is very likely to matter on a local level (as alluded to by the authors), but what about ozone formation on regional scales? In other countries, downwind of major cities, ozone formation transitions to NOx-sensitive due to the abundance of biogenic sources that can react alongside NOx (e.g. Trainer et al., 1987). I think this should be discussed as well, since NOx reductions matter and are important in the long run.

Minor Comments

Line 19: It would be good to note that the measurements at JAES were conducted using GC.

Lines 23-24: Awkward phrasing, recommend saying "We identified VOC sources using positive matrix factorization and assessed their contributions to photochemical O3 formation using an observation-based model employing the MCM"

Line 30: "control on" seems strong, given that other factors (e.g. meteorology) play a very important role. May suggest using "precursor to"

Line 32-33: Do you mean that the contribution of biogenic emissions to O3 was significantly lower than anthropogenic emissions? It would be useful to make this comparison.

Lines 45-48: The word "associated" suggests that rapid economic growth occurred because of increases in pollution. Would recommend replacing associated with "Rapid economic growth has led to ..."

Line 56: "VOCs" should be singular, since it is used as an adjective here. Other instances of this are found sparsely throughout the text.

Line 62: What do you mean by "industrial structure"? Does you mean that there is a high presence of industry in Ningbo?

Line 76: You could clarify here that you employ the entire MCM (v 3.2).

Line 78-79: Summarized, proposed, and assessed should be present tense here, since you are recommending these in the present manuscript

Line 100: When you say "the sample was enriched after 600 mL of air sample" do you mean "600 mL of air was sampled"? If so, the latter phrasing may be more clear.

Line 101: What is the "Dean's Switch" technology?

Line 107: Was this a custom calibration standard, or a commercially available standard? If commercially available, it would be good to quote the manufacturer. If prepared in-house, are there uncertainties in the VOC mixture?

Line 245: "Identified" is a Confusing word choice, since you identified the sources, not the model! I would recommend changing to "Five VOC sources were resolved by PMF"

References

Trainer, M., Williams, E., Parrish, D. *et al.* Models and observations of the impact of natural hydrocarbons on rural ozone. *Nature* **329,** 705–707 (1987) doi:10.1038/329705a0

---

## Author Comment (AC1) · 21 Jan 2020

In this study, the authors conducted one-year VOC observation at an urban site in Nanjing. They analyzed the seasonal and diurnal characteristics of 56 VOCs as well as their sources using the PMF model. A box model together with a Master Chemical Mechanism (MCM) was used to identify the relationships between the contributions of VOC sources and the $O_3$ formation. The results were also compared with VOCs data from other Chinese megacities like Beijing, Guangzhou, and Shanghai. VOC have been well recognized to be responsible for the swift development of air pollution events since volatile organic compounds are key precursors of $O_3$ and secondary organic aerosols (SOA). However, the speciation and emission strength of these VOC have been demonstrated to be hard to acquire due to the fact that VOC can be emitted from a diversity of domestic and industrial activities. Therefore, measurements of VOC emissions are critically needed in China. This work can be a significant contribution to the atmospheric research community. Overall, the manuscript is fairly well written and I would recommend the manuscript for publication after minor revisions.

Reply: We highly appreciate the reviewer's positive comments and helpful suggestions. We have addressed all the comments/suggestions in the revised manuscript. Detailed responses to the individual specific comment/suggestion are as follows.

**Specific concerns:**

1. Line 61: What's "photochemical industry"?

Reply:Sorry for the mistake. It should be "petrochemical industry". For details, please refer to Line 62, page 2 in the revised manuscript.

2. Sample location: there is always an asphalt waterproof layer on the rooftop of an office building. How to avoid the interference of this emissions?

Reply: We thank the reviewer's valuable comment. There is a waterproof layer on the rooftop of the building but there was no guarantee that it was made of asphalt. Furthermore, despite this waterproof layer on the rooftop of the building, the interferences of emissions from this layer were believed to be insignificant because: 1) The waterproof layer was covered by the layer of concrete, which was further covered with a layer of ceramic tile; 2) The building had been built for three years before the sampling campaign was started; 3) It was documented that the VOC emitted from asphalt mainly included benzene, toluene, ethylbenzene and xylene. However, the levels of benzene, toluene, ethylbenzene, m/p-xylene and o-xylene were lower than those observed in other urban, industrial and rural environments in different regions (section 3.1, He et al., 2019; Mo et al., 2015, 2017; An et al., 2014 and 2015; Zhang et al., 2012). 4) The sampling inlet was about 2-3 m above the rooftop of the building. Indeed, there is a waterproof layer on the rooftop of building. Therefore, we believe that though there is a waterproof layer for the rooftop of the building, the interference of its emission on ambient VOCs was insignificant.

The above text has been added in the revised manuscript to highlight the insignificant influence of waterproof layer on the rooftop of the building

For details, please refer to Lines 88-96, Page 3 in the revised manuscript.

**Reference:**

Gardiner, M.S., Lange, C.R., 2005. Comparison of laboratory generated and field-obtained HMA VOCs with odour potential. The International Journal of Pavement Engineering 6, 257-263.

He Zhuoran, et al., 2019. Contributions of different anthropogenic volatile organic compound sources to ozone formation at a receptor site in the Pearl River Delta region and its policy implications. Atmospheric Chemistry and Physics 19, 8801-8816.

Zhang, Y.L., Wang, X.M., Blake, D.R., Li, L.F., Zhang, Z., Wang, S.Y., Guo, H., Lee, F.S.C., Gao, B., Chan, L.Y., Wu, D., Rowland, F.S., 2012. Aromatic hydrocarbons as ozone precursors before and after outbreak of the 2008 financial crisis in the Pearl River Delta region, south China. Journal of Geophysical Research 117, D15306, doi:10.1029/2011JD017356.

3. Aromatics are important of gasoline. So, source 5 could be identified as gasoline cars. Thus, the identification of source 5 to industrious emissions may be need more relevant tracers.

Reply: The reviewer's valuable comment is highly appreciated. We agreed with the reviewer that aromatic hydrocarbons, especially benzene, toluene, ethylbenzene and xylenes could also emitted from gasoline vehicles. However, in addition to gasoline vehicle emissions, industrial emission could be another important contributor to ambient aromatic hydrocarbons in the Yangtze River Delta, Pearl River Delta and North China Plain (Yuan et al., 2009; He et al., 2019; Zhang et al., 2013, 2014; Mo et al., 2015, 2017; An et al., 2014). The tunnel studies and emission-based measurement results found that aromatic hydrocarbons from gasoline vehicle exhaust were coherently emitted with pentanes, butenes, $n$-hexane, and cyclopentane, which were more consistent with the profile in source 3 mentioned above (Liu et al., 2008; Ho et al., 2009; Yuan et al., 2009; Zhang et al., 2018). Therefore, the absence of above species in source 5 indicated that this source could be related to industrial emission (Zhang et al., 2014). Particularly, the high presence of toluene, ethylbenzene, xylenes, ethyltoluene and trimethylbenzene was consistent with the emission-base measurement results conducted in paint and printing industries (Yuan et al., 2010) and manufacturing facilities (Zheng et al., 2013). On the other hand, the profile of high presence of aromatic hydrocarbons ($C_7$-$C_9$ aromatics) and the certain amount of ethene, was also agree with the profiles measured in the areas dominated by industrial emissions in the Yangtze River Delta region (An et al., 2014; Shao et al., 2016; Mo et al., 2017). For example, An et al. (2014) reported that toluene, ethylbenzene, xylenes, and trimethylbenzenes could be emitted from different industrial processes, and identified that the factors with high loadings of these species as industrial production, solvent usage and industrial production volatilization sources by PAC/APCS at the industrial

area in Nanjing. On the other hand, Mo et al. (2017) identified the factors with high concentrations of $C_7$-$C_9$ aromatics and ethene as residential solvent usage, chemical and paint industries and petrochemical industry with the PMF model applied to the data collected in an industrialized coastal city of Yangtze River Delta region. To further identify source 3 and source 5, the ratio of toluene/benzene (T/B, ppbv/ppbv) in each profile was compared with those obtained from emission-based measurements and tunnel study results (Zhang et al., 2018 and references therein). The ratios of T/B were ~8.2 and ~1.2 for sources 5 and 3, respectively, and were consistent with those of "industrial processes and solvent application", and "roadside and tunnel study", respectively (Zhang et al., 2018 and references therein). This further confirmed that source 3 was related to gasoline vehicular exhaust, while source 5 was associated with industrial emission.

The above discussion has been provided in the revised manuscript to further clarify source 5 and source 3. For details, please refer to Lines 432-456, Page 15 in the revised manuscript.

**References:**

Ho, K.F., et al., 2009. Vehicular emission of volatile organic compounds (VOCs) from a tunnel study in Hong Kong. Atmospheric Chemistry and Physics 9, 7491-7504.

Liu, Y., et al., 2008. Source apportionment of ambient volatile organic compounds in the Pearl River Delta, China: Part II. Atmospheric Environment 42, 6261-6274.

Shao, P., An, J.L., Xin, J.Y., Wu, F.K., Wang, J.X., Ji, D.S., Wang, Y.S., 2016. Source apportionment of VOCs and the contribution to photochemical ozone formation during summer in the typical industrial area in the Yangtze River Delta, China. Atmospheric Research 176-177, 64-74.

Yanli Zhang, et al., 2013. Species profiles and normalized reactivity of volatile organic compounds from gasoline evaporation in China. Atmospheric Environment 79, 110-118.

Yuan, B., Shao, M., Lu, S.H., Wang, B., 2010. Source profiles of volatile organic compounds associated with solvent use in Beijing, China. Atmospheric Environment 44, 1919-1926.

Zhang Yanli, et al., 2018. Decadal changes in emissions of volatile organic compounds (VOCs) from on-road vehicles with intensified automobile pollution control: case study in a busy urban tunnel in south China. Environmental Pollution 233, 806-819.

Zheng, J.Y., Yu, Y.F., Mo, Z.W., Zhang, Z., Wang, X.M., Yin, S.S., Peng, K., Yang, Y., Feng, X.Q., Cai, H.H., 2013. Industrial sector-based volatile organic compound (VOC) source profiles measured in manufacturing facilities in the Pearl River Delta, China. Science of the Total Environment 456-457, 127-136.

Zibing Yuan, et al., 2009. Source analysis of volatile organic compounds by positive matrix factorization in urban and rural environments in Beijing. Journal of Geophysical Research 114,

D00G15, doi: 10.1029/2008JD011190.

4. Why the diurnal trend of fuel evaporation showed a decrease at noon time since this source is temperature dependent?

Reply: Thanks a lot for the comment. Apart from emissions, ambient VOC concentrations are largely determined by photochemistry and dilution processes, in particular the variations of mixing height in the course of a day (Gillman et al., 2009; Wang et al., 2013). Though the evaporation of fuel is dependent on temperature, the average temperature in the morning and evening (*i.e,* 0800-1000 and 1700-1900 LT, respectively) when peaks of fuel evaporation were found was only about ~1.2 °C lower than that observed from noon to afternoon (1100-1600 LT), which may not result in much higher fuel evaporation at noon (the difference between maximum and minimum values for fuel evaporation was found to be ~6 $\mu g/m^3$). On the other hand, in addition to evaporation from the gas station, fuel could evaporate from hot engines, fuel tanks and the exhaust system when the car is running. Furthermore, the engine remains hot for a period of time after the car is turned off, and gasoline evaporation continues when the car is parked (Technology center, University of Illionois, https://mste.illinois.edu/tcd/ecology/fuelevap.html, access date: 25 December 2019). The similarity of diurnal variations of fuel evaporation to vehicular emissions suggested that the prominent peak in the morning and evening hours were related to the increased vehicles in the traffic rush hour and emissions accumulated in the relatively low boundary layer.

To provide detailed discussion on the diurnal pattern of fuel evaporation, the above analysis text has been added in the revised manuscript. For details, please refer to Lines 506-517, Pages 17-18 in the revised manuscript.

**References**

Gilman, J.B., Kuster, W.C., Goldan, P.D., Herndon, S.C., Zahniser, M.S., Tucker, S.C., Brewer, W.A., Lerner, B.M., Williams, E.J., Harley, R.A., Fehsenfeld, F.C., Warneke, C., de Gouw, J.A., 2009. Measurements of volatile organic compounds during the 2006 TexAQS/GoMACCS campaign: industrial influences, regional characteristics, and diurnal of dependencies of the OH reactivity. Journal of Geophysical Research 114, D7, doi: 10.1029/2008JD011525.

Wang, H.L., Chen, C.H., Wang, Q., Huang, C., Su, L.Y., Huang, H.Y., Lou, S.R., Zhou, M., Li, L., Qiao, L.P., Wang, Y.H., 2013. Chemical loss of volatile organic compounds and its impact on source analysis through a two-year continuous measurement. Atmospheric Environment 80, 488-498.

---

## Author Comment (AC2) · 21 Jan 2020

Zhao et al. describe VOC measurements conducted at the Jiangsu Academy of Environmental Science (JAES) in Nanjing, China. The authors measure VOCs using a GC system, and interpret the sources of these VOCs using positive matrix factorization. The authors evaluate the environmental impacts of these emissions on ozone formation using an observation-based model (OBM) employing the Master Chemical Mechanism (MCM v3.2), and identify the anthropogenic VOCs likely to be significant ozone precursors. The authors also evaluate ozone sensitivities to VOC and NOx reductions, and conclude that VOC reductions would be the best strategy to reduce ozone in Nanjing.

In general, the manuscript reads very well and is well-organized to tell a coherent message. I appreciate the authors work to carefully measure VOCs and benchmark these measurements against other cities in China. I am generally convinced by the PMF results given that the authors interpretation is reasonable, and the PMF factors are prescribed to obvious sources in the Nanjing area (which are very well described); however, I do have some recommendations that could improve the PMF analysis and strengthen the justification of source apportionment. Finally, I believe the use of the OBM is justified to evaluate VOC RIR, but I am not convinced that the OBM can be used to evaluate the ozone isopleth without further evidence that the model is doing an adequate job to capture ozone formation in the Nanjing region. My comments below

primarily address PMF and the OBM.

Reply: Thanks for the reviewer's positive comments and helpful suggestions. We have addressed all the comments/suggestions in the revised manuscript. Detailed responses to the individual specific comment/suggestion are as follows.

**Major comments**

1. The PMF solution appears to be reasonable; however, I believe the authors need to do more to show that the PMF solution is robust. In Section 2.2, the authors state that comparisons were made to observations, emissions inventories, and previous PMF analyses, but no evidence is shown here or in the supplement to convince the reader that this is true. Can the authors show the $Q/Q_{exp}$ and explain why they settled on a 5-factor solution? What was the factor space used? Did the authors vary other parameters (e.g. Fpeak) or conduct a bootstrapping analysis to estimate uncertainty? Can the authors show the comparisons to other factor profiles reported in literature (e.g. the industrial factor compared to An et al., 2014).

I ask because PMF is partly subjective, and a more thorough discussion is necessary to justify why the authors settle on the solution presented in the manuscript. A 5-factor solution seems reasonable, and the factors discussed all appear to be consistent with the sources surrounding the sampling site, but this could be shown with more evidence in the main text or supplemental information.

Reply: The reviewer's valuable comment is highly appreciated. To provide more evidence for the selection of the five-factor solution from the PMF mode, the following text has been added in the revised manuscript:

"*The PMF model was tested using a variety of factor numbers, and the optimum source profiles and contributions were determined based on the correlation between modelled and observed data, the comparison of modelled profiles with the results from emission-based measurements, and previous studies involving PMF/other receptor model simulations (i.e., HKEPD, 2015; Wang et al., 2014; An et al., 2014; Liu et al., 2008a). For example, different solution with different factor numbers was explored and the source apportionment results from a five-factor resolution that could sufficiently*

*explain the observed levels of VOCs were selected (details in Section 3.3). Compared with five-factor solution, the four-factor solution derived two profiles that attributable to gasoline and diesel vehicular exhaust, while most of the aromatic species in these sources and certain amounts of $C_3$-$C_4$ species from fuel evaporation were categorized under industrial emission. On the other hand, the six-factor solution has split a factor with high presence of ethyne and certain amounts of ethane (30% in species total), $C_3$ species and benzene (~20% in species total), while some alkenes (18-80% in species total) were incorporated into fuel evaporation. Furthermore, the performance of the five-factor solution was evaluated using various checks and sensitivity tests. Suitable correlations between the observed concentrations and those of each species predicted by the model were observed, with the correlation coefficients ($R^2$) ranging from 0.60 - 0.91, indicating that the solution adequately reproduced the observed variations of each species. All the scale residuals were within ± 3σ with normal distributions for all species (Baudic et al., 2016). Moreover, different numbers of start seeds were tested during the simulation and no-multiple solutions were found. The ratio of Q(robust)/Q(true) obtained was ~0.93, close to 1 as suggested by previous studies and the user guide manual (Lau et al., 2010; Ling et al., 2016; Paatero, 2000). In addition, the results from bootstrapping analysis for the five-factor solution with bootstrap random seed found that all the factors were mapped to a basic factor in all the 20 bootstrap runs, while the uncertainties of each species from bootstrapping analysis were within the range of 1~20%. In this study, different $F_{peak}$ values ranging from -5 to 5 was tested in the 5-factor solution for a more realistic profile (Lau et al., 2010; Baudic et al., 2016). The profiles with the nonzero $F_{peak}$ values were consistent with those with zero $F_{peak}$ value, reflecting that there was little rotation for the selected solution, confirming that the profiles were reasonably explained by the five-factor solution (Baudic et al., 2016). The results of $F_{peak}$ value = 0.5 (the base run) was selected for analysis in this study. Overall, the above features demonstrated that the five-factor solution from PMF could provide reasonable and stable apportionment results for the observed VOCs at the JAES site."*

For details, please refer to Lines 152-178, Pages 5-6 in the revised manuscript.

Furthermore, to justify the source apportionment results, more discussion based on the comparison of modelled profiles with the results from emission-based measurements, and other PMF model simulations were highlighted as follows:

[revised manuscript text omitted]

For details, please refer to Lines 404-456, Pages 14-15 in the revised manuscript.

**References:**

An, J.L., Zhu, B., Wang, H.L., Li, Y.Y., Lin, X., Yang, H., 2014. Characteristics and source apportionment of VOCs measured in an industrial area of Nanjing, Yangtze River Delta, China. Atmospheric Environment 97, 206-214.

Barletta, B., Meinardi, S., Rowland, F.S., Chan, C.Y., Wang, X.M., Zou, S.C., Chan, L.Y., Blake, D.R., 2005. Volatile organic compounds in 43 Chinese cities. Atmos. Environ. 39 (32), 5979-5990.

Baudic, A., Gros, V., Sauvage, S., Locoge, N., Olivier, S., Sarda-Esteve, R., Kalogridis, C., Petit, J.-E., Bonnaire, N., Baisnee, D., Favez, O., Albinet, A., Sciare, J., Bonsang, B., 2016. Seasonal variability and source apportionment of volatile organic compounds (VOCs) in the Paris megacity (France). Atmospheric Chemistry and Physics 16, 11961-11989.

He, Z.R., Wang, X.M., Ling, Z.H., Zhao, J., Guo, H., Shao, M., Wang, Z., 2019. Contributions of different anthropogenic volatile organic compound sources to ozone formation at a receptor site in the Pearl River Delta region and its policy implication. Atmospheric Chemistry and Physics 19, 8801-8816.

Ho, K.F., Lee, S.C., Ho, W.K., Blake, D.R., Cheng, Y., Li, Y.S., Ho, S.S.H., Fung, K., Louie, P.K.K., Park, D., 2009. Vehicular emission of volatile organic compounds (VOCs) from a tunnel study in Hong Kong. Atmospheric Chemistry and Physics 9, 7491-7504.

Lau, A.K.H., Yuan, Z.B., Yu, J.Z., Louie, P.K.K., 2010. Source apportionment of ambient volatile organic compounds in Hong Kong. Science of the Total Environment 408, 4138-4149.

Liu, Y., Shao, M., Lu, S.H., Chang, C.C., Wang, J.L., Fu, L.L., 2008. Source apportionment of ambient volatile organic compounds in the Pearl River Delta, China: Part II. Atmospheric Environment 42, 6261-6274.

Mo, Z.W., Shao, M., Lu, S.H., Niu, H., Zhou, M.Y., Sun, J., 2017. Characterization of non-methane hydrocarbons and their sources in an industrialized coastal city, Yangtze River Delta, China. Science of the Total Environment 593-594, 641-653.

Shao, P., An, J.L., Xin, J.Y., Wu, F.K., Wang, J.X., Ji, D.S., Wang, Y.S., 2016. Source apportionment of VOCs and the contribution to photochemical ozone formation during summer in the typical industrial area in the Yangtze River Delta, China. Atmospheric Research 176-177, 64-74.

Tasi, W.Y., Chan, L.Y., Blake, D.R., Chu, K.W., 2006. Vehicular fuel composition and atmospheric emissions in South China: Hong Kong, Macau, Guangzhou, and Zhuhai. Atmospheric Chemistry and Physics 6, 3281-3288.

Yuan, Z.B., Lau, A.K.H., Shao, M., Louie, P.K.K., Liu, S.C., Zhu, T., 2009. Source analysis of volatile organic compounds by positive matrix factorization in urban and rural environments in Beijing. Journal of Geophysical Research 114, D00G15, doi:10.1029/2008JD011190.

Zhang, Y.L., Wang, X., Barletta, B., Simpson, I.J., Blake, D.R., Fu, X., Zhang, Z., He, Q., Liu, T., Zhao, X., Ding, X., 2013. Source attributions of hazardous aromatic hydrocarbons in urban, suburban and rural areas in the Pearl River Delta (PRD) region. Journal of Hazardous Materials 250, 403-411.

Zhang, Y.L., Yang, W.Q., Simpson, I.J., Huang, X.Y., Yu, J.Z., Huang, Z.H., Wang, Z.Y., Zhang, Z., Liu, D., Huang, Z.Z., Wang, Y.J., Pei, C.L., Shao, M., Blake, D.R., Zheng, J.Y., Huang, Z.J., 2018. Decadal changes in emissions of volatile organic compounds (VOCs) from on-road vehicles with intensified automobile pollution control: case study in a busy urban tunnel in south China. Environmental Pollution 233, 806-819.

Zheng, J.Y., Yu, Y.F., Mo, Z.W., Zhang, Z., Wang, X.M., Yin, S.S., Peng, K., Yang, Y., Feng, X.Q., Cai, H.H., 2013. Industrial sector-based volatile organic compound (VOC) source profiles measured in manufacturing facilities in the Pearl River Delta, China. Science of the Total Environment 456-457, 127-136.

2. The authors employ an OBM to evaluate ozone sensitivity to VOCs and NO$_x$. OBMs are primarily useful because they allow one to evaluate relative incremental reactivity (as the authors describe in section 2.3). One strength of an OBM is that you do not need all of the measurements that described ozone formation; rather, you calculate source functions that explain residual effects on the time evolution of a measured species (e.g. meteorology, chemistry not accounted for in the mechanism, additional precursors that contribute to ozone formation, etc). From these calculations, you can derive the RIR by conducting a small perturbation on the system (e.g., decreasing or increasing the concentration of a species that is measured and well-represented by the model). The calculation of RIR are good and justified with the use of an OBM.

Reply: The reviewer's positive comment on OBM is highly appreciated. Yes, in this study, RIR which has been adopted in previous studies (i.e., Wang et al., 2017; Lyu et

al., 2016; Xue et al., 2014; Cheng et al., 2010) as used to assess the sensitivity of precursors to photochemical $O_3$ formation by changing the concentrations of precursors (i.e., 10% reduction).

**References**

Cheng, H.R., Guo, H., Wang, X.M., Saunders, S.M., Lam, S.H.M., Jiang, F., Wang, T., Ding, A., Lee, S., Ho, K.F., 2010. On the relationship between ozone and ite precursors in the Pearl Rive Delta: application of an observation-based model (OBM). Environmental Science and Pollution Research 17, 547-560.

Lyu, X.P., Guo, H., Simpson, I.J., Meinardi, S., Louie, P.K.K., Ling, Z.H., Wang, Y., Liu, M., Luk, C.W.Y., Blake, D.R., 2016. Effectiveness of replacing catalytic converters in LPG-fueled vehicles in Hong Kong. Atmospheric Chemistry and Physics 16, 6609-6626.

Wang, Y., Wang, H., Guo, H., Lyu, X.P., Cheng, H.R., Ling, Z.H., Louie, P.K.K., Simpson, I.J., Meinardi, S., Blake, D.R., 2017. Long-term O3-precursor relationships in Hong Kong: field observation and model simulation. Atmospheric Chemistry and Physics 17, 10919-10935.

Xue, L.K., Wang, T., Gao, J., Ding, A.J., Zhou, X.H., Blake, D.R., Wang, X.F., Saunders, S.M., Fan, S.J., Zuo, H.C., Zhang, Q.Z., Wang, W.X., 2014. Ground-level ozone in four Chinese cities: precursors, regional transport and heterogeneous processes. Atmospheric Chemistry and Physics 14, 13175-13188.

In Section 3.4, the authors extend this analysis to evaluate the ozone isopleth. In this context, I don't believe the use of an OBM is justified. Isopleth calculation are defendable if a large fraction of the local, photochemically produced ozone is explained by the measured precursors. If a significant fraction of this produced ozone is explained by the time-dependent source function (i.e, the "residual" ozone), then the authors may not be measuring (or including in the model) a significant fraction of the VOC precursors needed to derive ozone formation. In that case, how can the authors determine whether Nanjing is VOC or $NO_x$-sensitive? The isopleth presented in Fig.6 is very $NO_x$ saturated, which the authors say generally agrees with previous literature. But do the measurements really defend this?

Reply: Thanks a lot for the reviewer's comment. We agree with the reviewer that the isopleth calculation could be reasonable when photochemically produced ozone from the measured precursors made a significant contribution to observed $O_3$. As the $O_3$ simulated by the OBM was derived based on the observed mixing ratios of precursors and local meteorology, it is more appropriate to refer the simulation of $O_3$ by the OBM

as local produced $O_3$, though the observed mixing ratios of precursors could be both influenced by local emissions and those transported from upwind areas (Liu et al., 2019). Therefore, before using the isopleth calculation, it is necessary to investigate whether locally produced $O_3$ from OBM could make a significant portion to the observed $O_3$ at the JAES site. The investigation on the wind parameters found that the average wind speed was ~1.8 m/s on $O_3$ episode days, with about ~51% of most wind speed data being $\leq 2$ m/s (see Figure S2 in the supplementary), suggesting that regional transport may not have a significant influence on the levels of $O_3$ and its precursors on episode days at the JAES site (Shao et al., 2016; Wang et al., 2017).

Figure S3 in the supplementary showed the timeseries for the comparison between local produced $O_3$ from OBM and the observed $O_3$ mixing ratios, while Figure S4 in the supplementary presented the comparison between the mean diurnal variations of simulated and observed $O_3$ mixing ratios for all the $O_3$ episode days at the JAES site. It was found that the model underestimated or overestimated $O_3$ on some episode days. The comparison between the mean diurnal variations of simulated and observed $O_3$ suggested that the model captured the diurnal variations of $O_3$, and the predicted and observed maximum $O_3$ values were comparable, though the predicted mixing ratios of $O_3$ were lower than that observed in the early morning. The above discrepancy between observed and predicted $O_3$ mixing ratios was mainly due to the failure to consider physical processes (i.e., horizontal and vertical transport) and/or the other $O_3$ precursors (i.e., carbonyls and other oxygenated VOCs (OVOCs)) (Liu et al., 2019; Cheng et al., 2010; Shao et al., 2009a). However, the simulation indeed provided a reasonable description of the $O_3$ variations using the observation data. To assess the local photochemically produced $O_3$ against the measured levels of $O_3$ during the $O_3$ episode days, the amount of the locally produced $O_3$ formed by the photochemistry was compared with the observed $O_3$ accumulations that were calculated as difference between the peak and early-morning concentrations of $O_3$. The amount of local photochemically produced $O_3$ was determined by the net $O_3$ production rate, which was calculated by the difference between the gross production $G(O_3)$ and destruction rates $D(O_3)$ in the model (Equations 1-3).

$$P(O_3) = G(O_3) - D(O_3) \qquad (1)$$

$$G(O_3) = k_{HO_2+NO}[HO_2][NO] + \sum k_{RO_2i+NO}[RO_{2i}][NO] \qquad (2)$$

$$D(O_3) = k_{HO_2+O_3}[HO_2][O_3] + k_{OH+O_3}[OH][O_3] + k_{O(^1D)+H_2O}[O(^1D)][H_2O]$$
$$+ k_{OH+NO_2}[OH][NO_2] + k_{alkenes+O_3}[alkenes][O_3] \qquad (3)$$

Where the $k$ constant values were the rate coefficients for the subscript reaction. The detailed description for the above calculation was provided by Xue et al. (2013, 2014) and Wang et al. (2017). At JAES, the daytime (07:00–19:00 LT) average net $O_3$ production rate was estimated to be 6.2 ppbv h$^{-1}$, corresponding to ~74 ppbv $O_3$ formed from local photochemistry during daytime hours. The amount was coincident with the average increment of $O_3$ observed from early morning to late afternoon at JAES (~81 ppbv), suggesting that local photochemically produced $O_3$ significantly contributed to the $O_3$ increment at JAES. Indeed, the observed minimum $O_3$ mixing ratios before accumulation which were considered as the residual $O_3$ levels (or background $O_3$, the mean value was 20 ppbv) at the sampling site as suggested by the previous studies (Xue et al., 2013, 2014a,b) only accounted for ~20% of the observed maximum $O_3$ values (the mean value was 102 ppbv) (data not shown). Furthermore, the difference between observed and simulated minimum $O_3$ mixing ratios which was considered as the fraction of residual $O_3$ that could not be explained by the OBM model only contributed ~17% of the observed maximum $O_3$ mixing ratio. The above analysis on the difference between observed and simulated $O_3$ levels confirmed that local photochemical produced $O_3$ made a significant fraction to observed $O_3$ levels at JAES. However, we admitted that the OBM model could not accurately estimate the contributions of residual $O_3$ to the increment of $O_3$ values during daytime, which requires to be studies using a combination of different models and dataset (i.e., the regional air quality model, Lagrangian dispersion model and emission inventory) (Wang et al., 2015; Ding et al., 2013a, b; Jiang et al., 2010).

To further evaluate the model performance, the index of agreement (IOA) that was developed to assess the agreement between modelled and observed results was used in this study (Huang et al., 2005; Wang et al., 2013, 2015; Liu et al., 2019). The calculation

of IOA was as follows:

$$IOA = 1 - \frac{\sum_{i=1}^{n}(O_i - S_i)^2}{\sum_{i=1}^{n}(|O_i - \overline{O}| + |S_i - \overline{O}|)^2} \qquad (4)$$

Where $S_i$ and $O_i$ were the simulated and observed $O_3$, respectively, while $\overline{O}$ was the mean of observed $O_3$, and $n$ is the number of samples. The IOA values ranged between 0 and 1, and a relatively higher value of IOA indicated relatively greater consistency between simulated results and observation data (Wang et al., 2013, 2015, 2017). In this study, the IOA of $O_3$ was ~0.85, suggesting consistency of the abundance and variation of $O_3$ between the observation and simulation, and demonstrating that locally produced $O_3$ could be explained by the measured precursors.

In the atmosphere, the sensitivity of photochemical $O_3$ formation was distributed into three regimes, including the VOC-limited regime, the $NO_x$-limited regime and the transitional regime. In the VOCs-limited regime (the relative concentration $[NO_x]/[VOC]$ is high and/or $NO_x$ is saturated), photochemical $O_3$ formation decreases with the decrease in the concentration of VOCs (resulting from the control of VOC emissions), while in the $NO_x$-limited regime (high $[VOC]/[NO_x]$ ratio and/or VOC is saturated), any reduction in the $NO_x$ concentration would shortens the $O_3$ formation chain length and reduces the photochemical $O_3$ formation (Jenkin and Clemitshaw, 2000). The mean mixing ratios of $NO_x$ and TVOCs during daytime hours (0700-1800 LT, local time) on $O_3$ episode days were 19.2 ± 1.2 ppbv, respectively, with the mean ratio of VOCs/$NO_x$ as (ppbC/ppbv) ~3.4, suggested that the atmosphere in at the JAES site was $NO_x$ saturated and photochemical $O_3$ formation located in the VOC-regime (Jenkin and Clemitshaw, 2000). However, it should be noted that using the ratios of VOCs/$NO_x$ to determine the $O_3$ formation regime could be biased in different environments as different VOC species react at different rates and with different reaction mechanisms, thus inducing the nonlinear dependency of $O_3$ formation on $NO_x$ and VOCs. Figure 6 shows the $O_3$ isopleth plot illustrating the relationship between

VOCs and $NO_x$ concentrations on the $O_3$ mixing ratio. The plot is the output from the OBM-MCM model, and is based on the mean diurnal variability of observed air pollutants on $O_3$ episode days. Based on the current scenario (with 100% of observed mixing ratios of VOCs and $NO_x$, point A in Figure 6), the $O_3$ mixing ratio decreased with the reduction of VOCs and increased with the reduction of $NO_x$, indicating that $O_3$ formation in this site is VOC-limited. Furthermore, to accurately evaluate the $O_3$-precursor relationship, the RIR values from the OBM model, which were frequently used to evaluate the $O_3$ formation sensitivity based on observation data, were further explored. Positive RIR values were found for the VOCs (see Figure 7a), while negative values were found for $NO_x$ (i.e., -0.25 ± 0.02), further confirming that $O_3$ formation at the JAES site was VOC-limited (Zhang et al., 2008; Shao et al., 2009b; Cheng et al., 2010).

Overall, the above analysis confirmed that simulation of OBM could be used to conduct $O_3$ isopleth calculation and investigate the $O_3$-precursor relationship (Shao et al., 2016; Lyu et al., 2019; He et al., 2019).

The above evaluation on the OBM model performance has been provided in Lines 209-265, Pages 7-9 in the revised manuscript, while the $O_3$-precursor relationship evaluated by the observed data, $O_3$ isopleth plot and RIR values from the OBM-MCM model was provided in details in Lines 511-535, Page 18 in the revised manuscript.

Also, is there a reason that v 3.2 was used, rather than 3.3.1? v3.3.1 has updates to the isoprene mechanism that may (or may not) be relevant here.

Reply: Thank for the reviewer's comment. Compared with version 3.2, MCM version 3.3.1 have the following updates: 1) The updates of $HO_x$ recycling which has a collectively significant impact on OH radical regeneration at lower $NO_x$ levels. 2) The updates of $NO_x$ recycling which include the formation of species that have been reported to play a role in SOA-formation mechanisms, including epoxydiols (initially implemented in MCM v3.2), hydroxymethyl-methyl-a-lactone (HMML) and methacrylic acid epoxide (MAE). These above updates may not be relevant to the present study as the levels of $NO_x$ at the JAES site were not low, with the average mixing ratios of 25 ± 1 (mean ± 95% confidence interval) ppbv in 2016. Furthermore, the present study only focused on the roles of VOCs on $O_3$ formation through gas-phase mechanisms which may not be influenced by the $NO_x$ recycling updates for SOA formation. However, we did try our best to incorporate the version 3.3.1 MCM into our photochemical box model to evaluate the roles of NMHCs on SOA formation, which is still under construction (Ling et al., "Formation and sink of glyoxal and methylglyoxal

in a polluted subtropical environment: observation-based photochemical analysis and impact evaluation", unpublished manuscript).

Line 164: By TVOC, you mean the sum of measured VOCs?

Reply: Yes. It has been revised as follows:

"*The annual average total VOC (TVOC, sum of the measured VOCs)….*"

For details, please refer to Line 291, Page 9 in the revised manuscript.

Line 172: Is this reversed? The first number (referring to weekdays) is lower than the second (referring to weekends).

Reply: Sorry for the mistake. It has been revised accordingly. For details, please refer to Lines 298-299, Page 10 in the revised manuscript.

Table 1: You only give an average and standard deviation - no mixing ratio ranges are shown. I recommend removing "range".

Reply: Thanks for the reviewer's comment. The "range" has been removed accordingly. For details, please refer to Table 1 in the revised manuscript.

Line 194: Continuous VOC measurements have been available much longer than this in other countries. I would recommend changing this wording to say "online VOC measurements have been available for multiple decades"

Reply: Thanks for the suggestion. The sentence has been revised as "…online VOC measurements have been available for multiple decades…"

For details, please refer to Lines 319-320, Page 11 in the revised manuscript.

Fig.2 This is a nice benchmark of the Nanjing measurements with other cities during a period when developed countries were still reducing mobile emissions (mid 1990s – early 2000s). How does this compare with measurements conducted in developed countries today? It would be nice to see how the mixture in Nanjing compared to London or Los Angeles today, and would also highlight the gap that could be achieved

with further VOC reductions.

Reply: Thanks a lot for the reviewer's positive comment. In this study, to highlight the variations of VOCs in different regions, comparison of annual average concentrations of ambient VOCs in different cities based on real-time online continuous measurements of at least one year was present in Figure 2. We do try our best to find as many studies focusing on the long-term (at least one-year) variations of VOCs in developed regions/countries as possible. The results presented in Figure 2 were all the data we can get. It was found that the current ambient VOC concentrations in Chinese megacities are generally comparable to the urban VOC levels in developed countries during the year 2000. However, in developed countries, the mixing ratios of VOCs were observed to decrease in the recent decades following the implementation and formulation of VOC strategies (Warneke et al., 2012). For example, the mixing ratios of VOCs in Los Angeles have decreased significantly from 1960-2002 at an average annual rate of ~7.5%, while the mixing ratios of VOCs in London presented a higher and faster decreased since 1998 when there were higher VOC mixing ratios than those in Los Angeles, confirming that the earlier implementation of VOC reduction strategies in California had clearly led to the earlier improvement of air quality compared to London (Warneke et al., 2012; von Schneidemesser et al., 2010). Chinese megacities are therefore experiencing significnatly higher ambient VOCs contamination, given the remarkable decrease in VOC emissions in developed countries over the last two decades (European Environment Agency, 2016; U.S. EPA, 2017; Pan et al., 2015). High VOC levels in Chinese megacities are known to impact ambient ozone and secondary particle pollution, as well as cause adverse impacts on human health. However, as China has a solid foundation for VOCs monitoring and control, numerous strict, appropriate and targeted reduction strategies for VOCs have been/are being formulated and implemented in Chinese megacities (Guo et al., 2017). It is expected these measures could help China to reduce VOC emissions/mixing ratios and improve air quality in the future.

To highlight the reduction of VOCs in developed regions and the gap that could be achieved with further VOC reductions in China, the above discussion has been added

in the revised manuscript. For details, please refer to Lines 332-346, Pages 11-12 in the revised manuscript.

Figure 6. The isopleth description is somewhat confusing-is this % change in NOx and VOCs, or % of base-case VOCs.

Reply: Sorry for the mistake and confusion it caused. The horizontal and vertical axes in Figure 6 correspond to the percentage of base-case VOCs and $NO_x$. It has been revised accordingly in the caption of Figure 6.

Section 3.5. Without more work to convince the reader that the ozone isopleth is reasonable, I believe these statements would need to be amended. First, the authors haven't shown that the ozone precursors measured account for the majority of the ozone modeled in the OBM. Second, the recommendation to prioritize VOC reductions (line

381) is very likely to matter on a local level (as alluded to by the authors), but what about ozone formation on regional scales? In other countries, downwind of major cities, ozone formation transitions to $NO_x$-sensitive due to the abundance of biogenic sources that can react alongside NOx (e.g. Trainer et al., 1987). I think this should be discussed as well, since NOx reductions matter and are important in the long run.

Reply: We highly appreciated for the reviewer's comment. As discussed previously, the simulation of OBM model could reproduce the observed variations of $O_3$ and the measured $O_3$ precursor indeed contributed a major fraction for the model $O_3$ from OBM and the observed $O_3$ mixing ratios. On the other hand, we agreed with the reviewer that the results in the present study were based on the local measurement conducted at the urban site of Nanjing city, which likely presented a local perspective. The recommendation to prioritize VOC reductions was likely matter to the urban area where $O_3$ formation was VOC-limited. However, $O_3$ pollution is a regional cross-boundary environmental issue rather than a local pollution problem. Apart from VOCs, $NO_x$ was another important precursor for $O_3$ formation with its dual roles in $O_3$ production (enhancing $O_3$ formation in non $NO_x$-saturated environment and titrating $O_3$ in $NO_x$-saturated environment). In other areas (i.e., the rural environment and/or the downwind areas of urban center in the same region) where the concentrations of $NO_x$ are low and/or there is a non $NO_x$-saturated environment, the situation may be different and controlling VOCs should be conducted cautiously (Ou, et al., 2016; Yuan et al., 2013; Zheng et al., 2010). Therefore, from a regional perspective, the benefits of VOCs control measures could be further evaluated with those of $NO_x$ (i.e., the appropriate ratios of VOC/$NO_x$ for the reduction of $O_3$ pollution) as well as the associated $O_3$-VOCs-$NO_x$ sensitivity. Therefore, one important concern for the policy formulation and implementation system is whether controlling VOCs and $NO_x$ individually or controlling both VOCs and $NO_x$ is more effective and appropriate for alleviating $O_3$ pollution. It is necessary to consider the reduction ratios of VOC/$NO_x$ when VOCs and $NO_x$ are simultaneously controlled. Finally, long-term monitoring studies are necessary to determine the cost-benefits and performance of each policy. To provide more accurate discussion on the controlling VOCs and $NO_x$ in a regional/local perspective,

the following text has been added:

"*Last but not the least, though the present study suggested that reducing VOC emissions could be more effective in controlling $O_3$ pollution in the urban area of Nanjing where photochemical $O_3$ formation was VOC-limited, the results were based on local measurements, which likely presented a local perspective. However, $O_3$ pollution is a regional cross-boundary environmental issue rather than a local pollution problem. Apart from VOCs, $NO_x$ was another important precursor for $O_3$ formation with its dual roles in $O_3$ production (enhancing $O_3$ formation in non $NO_x$-saturated environment and titrating $O_3$ in $NO_x$-saturated environment). In other areas (i.e., the rural environment and/or the downwind areas of urban center in the same region) where the concentrations of $NO_x$ are low and/or there is a non $NO_x$-saturated environment, the situation may be different and controlling VOCs should be conducted cautiously (Ou, et al., 2016; Yuan et al., 2013; Zheng et al., 2010). Therefore, from a regional perspective, the benefits of VOCs control measures could be further evaluated with those of $NO_x$ (i.e., the appropriate ratios of VOC/$NO_x$ for the reduction of $O_3$ pollution) as well as the associated $O_3$-VOCs-$NO_x$ sensitivity. Therefore, one important concern for the policy formulation and implementation system is whether controlling VOCs and $NO_x$ individually or controlling both VOCs and $NO_x$ is more effective and appropriate for alleviating $O_3$ pollution. It is necessary to consider the reduction ratios of VOC/$NO_x$ when VOCs and $NO_x$ are simultaneously controlled. Finally, long-term monitoring studies are necessary to determine the cost-benefits and performance of each policy.*"
For details, please refer to Lines 653-668, Page 23 in the revised manuscript.

Minor Comments

Line 19: It would be good to note that the measurements at JAES were conducted using GC.

Reply: Thanks for the reviewer's suggestion. To highlight we use GC for the measurements, it has been revised as:

"*we conducted a one-year sampling exercise using a thermal desorption-GC (gas chromatography) system …..*"

Lines 23-24: Awkward phrasing, recommend saying "We identified VOC sources using positive matrix factorization and assessed their contributions to photochemical O3 formation using an observation-based model employing the MCM".

Reply: Thanks for the comment. The sentence has been revised accordingly. For details, please refer to Lines 23-25, Page 1 in the revised manuscript.

Line 30: "control on" seems strong, given that other factors (e.g. meteorology) play a very important role. May suggest using "precursor to"

Reply: Thanks for the suggestion. We have revised it accordingly. For details, please refer to Line 30, Page 1 in the revised manuscript.

Line 32-33: Do you mean that the contribution of biogenic emissions to O3 was significantly lower than anthropogenic emissions? It would be useful to make this comparison.

Reply: Yes, by considering both the reactivity and abundance of VOC species, the contribution of biogenic emissions to $O_3$ pollution was significantly lower than anthropogenic emissions. The text has been revised as follows:

"….. *the contribution of biogenic emissions to $O_3$ pollution was significantly reduced and lower than vehicular and industrial emissions.*"

For details, please refer to Lines 33-34, Page 1 in the revised manuscript.

Lines 45-48: The word "associated" suggests that rapid economic growth occurred because of increases in pollution. Would recommend replacing associated with "Rapid economic growth has led to ..."

Reply: Thanks for the suggestion. The text has been revised accordingly. Please refer to Lines 46-47, Page 2 in the revised manuscript.

Line 56: "VOCs" should be singular, since it is used as an adjective here. Other

instances of this are found sparsely throughout the text.

Reply: Thanks for pointing this out. It has been revised accordingly in the text. Furthermore, other instances have been double-checked and corrected.

Line 62: What do you mean by "industrial structure"? Does you mean that there is a high presence of industry in Ningbo?

Reply: Yes. Ningbo is a coastal city located on the southern wing of the Yangtze River Delta with a high presence of petrochemical industry. Since petrochemical industry is the leading industry in Ningbo, the results of source apportionment show that petrochemical industry is the main source of VOCs in Ningbo (Mo et al., 2015, 2016). The description for industries in Ningbo has been added as followed:

"……*which is a coastal city located on the southern wing of the Yangtze River Delta with petrochemical industry as its lead industry (Mo et al., 2015, 2016).*"

For details, please refer to Lines 63-64, Page 2 in the revised manuscript.

Line 76: You could clarify here that you employ the entire MCM (v 3.2).

Reply: Thanks for pointing this out. It has been revised in the manuscript accordingly (Line 79, Page 3)

Line 78-79: Summarized, proposed, and assessed should be present tense here, since you are recommending these in the present manuscript.

Reply: Yes. It has been revised accordingly (Lines 81-82, Page 3).

Line 100: When you say "the sample was enriched after 600 mL of air sample" do you mean "600 mL of air was sampled"? If so, the latter phrasing may be more clear.

Reply: Thanks a lot for the comment. To clarify the sample collection, the text have been revised as follows:

"*The sampling flow was 15 mL/min. After 600 mL of air was sampled, the cold trap was heated to resolve the compounds adsorbed on to it.*"

For details, please refer to Lines 119-120, Page 4 in the revised manuscript.

Line 101: What is the "Dean's Switch" technology?

Reply: Thanks for the comment. Dean's Switch technology is the technology that transfers the effluent from one column to another column with a different stationary phase. By this technology, all co-eluting impurities in the target analyte and the transferred peak are completely eluted.

To introduce this technology, the following description was added:

"……*By applying the Dean's Switch technology whereby the technology that transfers the effluent from one column to another column with a different stationary phase,……*"

For details, please refer to Lines 120-121, Page 4 in revised manuscript.

Line 107: Was this a custom calibration standard, or a commercially available standard? If commercially available, it would be good to quote the manufacturer. If prepared in-house, are there uncertainties in the VOC mixture?

Reply: Thanks for pointing this out. The VOC standard was purchased commercially. The following description was added:

"*Seven analyses were performed repeatedly to test the precision of the 56 species. Calibrant concentrations in the gas standard mixture (56 $C_2$-$C_{12}$ NMHCs, Linde Spectra Environment Gases, Inc, USA) ranged from 20 to 49 ppbC.*"

For details, please refer to Lines 128-129, Page 4 in the revised manuscript.

Line 245: "Identified" is a confusing word choice, since you identified the sources, not the model! I would recommend changing to "Five VOC sources were resolved by PMF".

Reply: Thanks for the reviewer's recommendation. It has been revised as suggested in Line 388, Page 13.

---

## Referee Report (RR1)

Zhao et al. present a revised manuscript describing the VOC measurements and ozone formation observed in Nanjing, China. I appreciate the authors response to my suggestions of how to improve the PMF solution. The additional details describing how the authors validated the PMF solution and the expanded discussion of PMF factor assignments are much improved.

However, I do not believe the authors have demonstrated that the OBM is reasonably capturing local ozone formation; therefore, I don't believe it is justifiable to use the model for ozone isopleth calculations. The authors have presented two new figures to compare the OBM with ozone observations (Figs. S3-S4) and text discussing these results (Lines 220-266). The comparison between the OBM and observed ozone concentrations (Fig. S3) is difficult to see, but in general, it seems that the model over or under predicts ozone by a factor of 2 on any given day. The authors acknowledge a number of shortcomings of the model (not capturing meteorological conditions, not capturing transported ozone, missing precursors, etc. line 227), which may explain many of the disagreements. I do not expect the authors to capture all of the ozone features over the entire sampling period (e.g. at night or during weather events); however, the model should capture daytime ozone production, especially during the ozone episodes defined in Fig S7. The model disagreements average out to a diurnal pattern that appears to be successful (Fig. S4), and the authors use this diurnal pattern to argue that the model is successful at recreating ozone production rates. This discussion is misleading given the results from Fig. S3.

As written, I don't believe the OBM should be included in this paper. Significant work would be needed to improve the OBM (see suggestions below). The measurements and PMF results are useful, and I encourage the authors to focus on these. Furthermore, I believe the authors could still address the importance of VOC precursors in ozone formation by evaluating proxies such as OH reactivity or maximum incremental reactivity (MIR).

Suggestions for OBM:

(1) The authors define an episode based on periods when ozone exceeds 80 ppb (I assume this is hourly averaged?). Based on Fig S7, this would suggest that the authors are comparing the OBM to data collected between April and October. I would be quite surprised if the boundary layer dynamics used by the authors apply equally to ozone episodes observed in April with those observed in October. Furthermore, it is not clear if adjustments were made to the TUV model to account for photon attenuation (e.g. clouds). Why not focus on a shorter period where the OBM can be tailored to the meteorological conditions measured over a week, as opposed to 5 months?

I would assume that the best period to choose would be (a) when winds are stable, slow, and originating from a single location (b) when skies are clear, and (c) when the boundary layer height can be modeled, measured, or well-represented by the approximation described by the authors.

(2)  The authors initialize each episode event with a spin-up period of two days that uses the campaign-averaged diurnal profile of each measurement. There seems to be a lot of variability in the monthly concentrations of VOCs, ozone, and NOx (Fig S5-S7). Why not use the hourly data and constrain each event to the measurements conducted each day? Since this analysis is focused on local ozone production, it seems that this would also help to account for ozone transported from upwind sources.

(3)  While the focus is on ozone formation, it's also important that the model should reasonably represent NOx and VOC profiles during ozone episodes. This not only affects radical budgets, but is also important in order to differentiate between ozone formed via reactions of NOx alongside biogenic and anthropogenic VOCs. It would be convincing to see how the model performs in reproducing VOC and NOx concentrations.

---

## Author Response (AR2)

**Response to Reviewer**

We appreciate the reviewer for the constructive and valuable comments, which were of great help in improving the quality of the manuscript. We have revised the manuscript accordingly and our detailed responses are shown below. All the revision is highlighted in the revised manuscript.

Zhao et al. represent a revised manuscript describing the VOC measurements and ozone formation in Nanjing, China. I appreciate the authors response to my suggestions of how to improve the PMF solution. The additional details describing how the authors validated the PMF solution and the expanded discussion of PMF factor assignments are much improved.

Reply: Many thanks for the reviewer's positive comments on the improvement of PMF simulation and source apportionment results in the manuscript.

However, I do not believe the authors have demonstrated that the OBM is reasonably capturing local ozone formation; therefore, I don't believe it is justifiable to use the model for ozone isopleth calculations. The authors have presented two new figures to compare the OBM with ozone observations (Figs. S3-S4) and text discussing these results (Lines 220-266). The comparison between the OBM and observed ozone concentrations (Fig. S3) is difficult to see, but in general, it seems that the model over or under predicts ozone by a factor of 2 on any given day. The authors acknowledge a number of shortcomings of the model (not capturing meteorological conditions, not capturing transported ozone, missing precursors, etc. line 227), which may explain may of the disagreements. I do not expect the authors to capture all of the ozone features over the entire sampling period (e.g. at night or during weather events); however, the model should capture daytime ozone production, especially during the ozone episodes defined in Fig. S7. The model disgreements average out to diurnal pattern that appears to be successful (Fig. S4), and the authors use this diurnal pattern to argue that the model is successful at recreating ozone production rates. This discussion is misleading given the results from Fig. S3.

As written, I don't believe the OBM should be included in this paper. Significant work would be needed to improve the OBM (see suggestions below). The measurements and PMF results are useful, and I encourage the authors to focus on these. Furthermore, I believe the authors could still address the importance of VOC precursors in ozone formation by evaluating proxies such as OH reactivity or maximum incremental reactivity (MIR).

Reply: The reviewer's suggestion is highly appreciated. We agreed with the reviewer that the OBM model in this study could not present more accurate description on the $O_3$ variations at the JAES sites as: 1) The vertical transport and horizontal transport

were no considered in the model; 2) Some parameters, i.e., the variations of boundary layers, which were obtained from the reanalysis results in China (0.75º × 0.75º) with limited daytime hours (Guo et al., 2016), which could not represent the real boundary layers at the JAES site. On the other hand, the cloudiness, which could influence the solar radiation and albedo was not measured in this study. Therefore, there were still uncertainties for the simulation of photolysis rates from TUV based on the sampling time, longitude and latitude of the sampling site, and the default configuration of clouds and albedo, though the photolysis rate from the TUV model in the present study could provide reasonable estimation on the photolysis rates compared with observations in other areas (Wang et al., 2019; Li et al., 2011). 3) Some precursors, i.e., carbonyl compounds, were not measured in the present study. 4) Dry deposition, which was not measured in the present study and was configured as previous studies (Xue et al., 2014; Zhang et al., 2003). Therefore, according to the reviewer's suggestion, we deleted the analysis using the OBM model.

In addition, to evaluate the contributions of VOC sources and species in different sources to $O_3$ pollution, the $O_3$ formation potential (OFP) of these sources and species were determined by maximum incremental reactivity (MIR) method. The discussion on the OFP of different VOC sources and species were provided in the revised manuscript as follows:

"As important $O_3$ precursors, information on the contributions of VOCs sources and related species to $O_3$ formation is necessary for the formulation and implementation of VOC control measures. To achieve this goal, the Maximum Incremental Reactivity (MIR) method, which evaluates the $O_3$ formation potential (OFP) on the basis of mass concentrations and maximum incremental reactivities of VOCs of the OH radical, were adopted in the present study (Shao et al., 2009b, 2011; Mo et al., 2017). Figure 6 presented the relative contributions of individual VOC sources and related VOC species from PMF to OFP at the JAES site. Industrial emissions was found to have the largest contribution to OFP at JAES due to the high loadings of aromatic VOC species that have relatively high OH reactivities in this source profile (Atkinson and Arey, 2003),

with the OFP value of ~43 µg/m³ and the contribution percentage of ~32% to the total OFP of all sources, followed by diesel vehicular exhausts (~36 µg/m³, ~27%), gasoline vehicular exhausts (~32 µg/m³, ~24%), fuel evaporation (~13 µg/m³, ~10%) and biogenic emissions (~9 µg/m³,~7%) though the MIR value of isoprene was much higher than other species. Similarly, using the same method to evaluate OFP of different VOC sources, Mo et al. (2017) found that industrial emissions (including the emissions of petrochemical industry, chemical and paint industries, solvent usage) and vehicular emissions were the dominant VOC sources for the total OFP in an industrialized coastal city (i.e., Ningbo) in the YRD region. Therefore, our results further demonstrated the need to minimize VOC emissions from industrial emissions and vehicle exhausts in order to lower $O_3$ formation and photochemical pollution in YRD.

[Figure]

**Figure 6. (a) The contribution of individual source to the total OFP of all sources extracted from PMF and (b) OFP values of the top 10 VOC species in the different source categories.**

Based on the mass concentrations of individual species in each source, we found that *m,p*-xylene and toluene in industrial emissions and gasoline vehicular emissions, propene, ethene, toluene and *m,p*-xylene in diesel vehicular emissions, and *o*-xylene, 1,2,4-trimethylbenzene and ethene in industrial emissions to be the dominant species from VOC emissions contributing to photochemical $O_3$ formation. Thus, only a small number of VOC species can be monitored for the effective control of $O_3$ formation.

For details, please refer to Lines 396-422, Pages 14-15 in the revised manuscript.

(1) The authors define an episode based on periods when ozone exceeds 80 ppb (I assume this is hourly averaged?). Based on Fig S7, this would suggest that the authors are comparing the OBM to data collected between April and October. I would be quite surprised if the boundary layer dynamics used by the authors apply equally to ozone episodes observed in Arpil with those observed in October. Furthermore, it is not clear if adjustments were made to the TUV model to account for photon attenuation (e.g. clouds). Why not focus on a shorter period where the OBM can be tailored to the meteorological conditions measured over a week, as opposed to 5 months?

I would assume that the best period to choose would be a) when winds are stable, slow, and originating from a single location (b) when skies are clear, and (c) when the boundary layer height can be modeled, measured, or well-represented by the approximation described by the authors.

Reply: Thanks for the reviewer's comment. We agreed with the reviewer that the boundary layer dynamics was different in different months. In the present study, the configuration of boundary layer height was based on the reanalysis data in China with the spatial resolution of 0.75º × 0.75º reported by Guo et al. (2016). According to Guo et al. (2016), the output for the boundary layer height from reanalysis data in China was categorized into spring, summer, autumn and winter. The episode days (i.e., days with maximum hourly average mixing ratio of 80 ppbv during daytime) were selected from April to October because VOCs data were not collected from 03/11/2016 to 20/11/2016 due to the maintenance for the GC system. Therefore, the average conditions of boundary layer height in spring, summer and autumn in Guo et al. (2016) were selected for the episode days in this study. For example, the average boundary layer heights in the morning (0800 LT), in the afternoon (1400 LT) and at night (2000 LT) were within the ranges of 0.25-0.40, 1.2-1.6 and 0.2-0.60 km in spring, summer and autumn,

respectively. Therefore, for the model simulation, the configuration of boundary layer heights from 0.3 km to 1.5 km was reasonable. To investigate the uncertainties for the variation of boundary layer heights, sensitive analysis with variations of boundary layer heights (i.e., from 0.2-1.2 km and 0.3-1.6 km based on the above ranges) were conducted. It was found that the uncertainties for the variations in boundary layer height for the modelled $O_3$ mixing ratios were < 4% (data not shown). Consistently, the sensitivity analysis on the variations of boundary heights suggested that the variations of boundary layer height on the modelling results was negligible (i.e., < 3% in net $O_3$ production rates) in four cities (i.e., Beijing, Shanghai, Guangzhou and Lanzhou) in China (Xue et al., 2014). However, uncertainties still existed for the configuration of boundary layer height in this study as 1) the boundary layer heights were not measured in this study; 2) the spatial resolution for the reanalysis data was $0.75^o \times 0.75^o$, which could not represent the real conditions at JEAS site.

On the other hand, according to the reviewer's suggestion, we selected the $O_3$ episode days when (a) when winds are stable, slow, and originating from a single location (b) when skies are clear and (c) the IOA between the simulation and observation > 0.9 (the IOA ranged within 0.70~0.92 for the whole $O_3$ episode days). In total, 11 days were selected as the following figure (Figure 1). However, it was found that the model still overpredicted or underpredicted the mixing ratios due to the factors mentioned above, though the $O_3$-precursor relationship, the contributions of each sources to $O_3$ formation on these 11 selected days were similar to those on all the $O_3$ episode days. In addition, a complete picture for all the $O_3$ episode days could not be provided if only 11 episode days were selected.

[Figure]

Figure 1 The comparison between observation and simulation results in days with the IOA > 0.90.

Therefore, as suggested by the reviewer above, we deleted the analysis using OBM-

MCM model, and used the MIR method to evaluate the contributions of VOC sources and species in different sources to $O_3$ pollution.

(2) The authors initialized each episode event with a spin-up period of two days that uses the campaign-average diurnal profile of each measurement. There seems to be a lot of variability in the monthly concentrations of VOCs, ozone and NOx (Figs S5-S7). Why not use the hourly data and constrain each event to the measurements conducted each day? Since the analysis is focused on local ozone production, it seems that this would also help to account for ozone transported from upwind sources.

Reply: Thanks for the reviewer's valuable comment and Sorry for the inappropriate description in the manuscript. Indeed, we used observed hourly data as the model input for the spin-up simulation if the observation data on the spin-up days were available. On the other hand, for days that not all the observation data were available, the spin-up simulation was conducted using the monthly averaged diurnal profiles of observation data.

(3) While the focus is on ozone formation, it's also important that the model should reasonably represent NOx and VOC profiles during ozone episodes. This not only affects radical budgets, but is also important in order to differentiate between ozone formed via reactions of NOx alongside biogenic and anthropogenic VOCs. It would be convincing to see how the model performs in reproducing VOC and NOx concentrations.

Reply: Many thanks for the reviewer's comment. As mentioned in the manuscript, though the model simulation was conducted using observation data as input, the evolution of VOCs and $NO_x$ in the model might not be the same as in the real atmosphere. To investigate the model performance in simulating VOCs and $NO_x$, the correlation and IOA between observed and simulated VOCs or $NO_x$ during daytime hours (0600-1900 LT) were explored. As there were many species of VOCs, here we presented the correlation of total concentrations of VOCs between observation and

simulation. It was found that the simulated mixing ratios of NO, NO$_2$ and VOCs correlated well with those observed at the JAES site (Figure 2), with the correlation coefficients R$^2$ (IOA) as 0.65 (0.89), 0.60 (0.74) and 0.86 (0.93), suggesting that the model indeed provide a reasonable description for the simulation of VOCs and NO$_x$ at the JAES site.

[Figure]

Figure 2 The correlation between observed and simulated mixing ratios of NO, NO$_2$ and VOCs during daytime hours (0600-1900 LT, local time).

---

## Author Response (AR3)

**Responses to Editor**

We appreciate the Editor for his valuable comments that were of great help in improving the quality of the manuscript. We also thank the Editor for his patience and help during the revision of the manuscript.

Comments to the Author:

Please copy-edit the final version. There are still a couple of misspellings in the submitted manuscript.

Reply: The Editor's comments are highly appreciated. We have revised the manuscript accordingly and our detailed responses are shown below. All the revision is highlighted in the revised manuscript.

Line 396: ... of VOC sources...

Reply: It has been corrected. For details, please refer to Line 396, Page 14 in the revised manuscript.

Line 399: ... maximum incremental reactivities of VOCs with the OH radical...

Reply: It has been revised. For details, please refer to Line 399, Page 14 in the revised manuscript.

Line 401: Industrial emissions were found...

Reply: It has been corrected. For details, please refer to Line 401, Page 14 in the revised manuscript.

Line 401 cc: (you might want to rephrase this sentance or split it up in two - it is a very long sentence and ends up being grammatically wrong).

Reply: Thanks for the comment. The sentence has been revised as followed:

"Industrial emissions were found to have the largest OFP at JAES due to the high loadings of aromatic VOC species that have relatively high OH reactivities in this source profile (Atkinson and Arey, 2003), with the OFP value of ~43 $\mu g/m^3$ and the

contribution percentage of ~32% to the total OFP of all VOC sources, followed by diesel vehicular exhausts (~36 μg/m$^3$, ~27%), gasoline vehicular exhausts (~32 μg/m$^3$, ~24%) and fuel evaporation (~13 μg/m$^3$, ~10%). Furthermore, though the MIR value of isoprene was much higher than other VOC species, biogenic emissions only contributed ~7% (~9 μg/m$^3$) to the total OFP of all VOC sources as the relatively low mixing ratio of isoprene at the JAES site."

For details, please refer to Lines 401-408, Pages 14-15 in the revised manuscript.

Line 409: ... dominant VOC sources of the total....

Reply: It has been corrected. For details, please refer to Line 410, Page 15 in the revised manuscript.

figure 6: ...The contribution of individual sources to...

Reply: It has been corrected. For details, please refer to Figure 6 in the revised manuscript.